# Neuronal miR-9 promotes HSV-1 epigenetic silencing and latency by repressing Oct-1 and Onecut family genes

Yue Deng[1,2,3], Yuqi Lin[1,2,3], Siyu Chen[1,2,3], Yuhang Xiang[1,2,3], Hongjia Chen[1,2,3], Shuyuan Qi[1,2,3], Hyung Suk Oh[4], Biswajit Das ●[4,5], Gloria Komazin-Meredith[5,6], Jean M. Pesola[5], David M. Knipe ●[4], Donald M. Coen ●[5] & Dongli Pan ●[1,2,3] ✉

Herpes simplex virus 1 (HSV-1) latent infection entails repression of viral lytic genes in neurons. By functional screening using luciferase-expressing HSV-1, we identify ten neuron-specific microRNAs potentially repressing HSV-1 neuronal replication. Transfection of miR-9, the most active candidate from the screen, decreases HSV-1 replication and gene expression in Neuro-2a cells. Ectopic expression of miR-9 from lentivirus or recombinant HSV-1 suppresses HSV-1 replication in male primary mouse neurons in culture and mouse trigeminal ganglia in vivo, and reactivation from latency in the primary neurons. Target prediction and validation identify transcription factors Oct-1, a known co-activator of HSV transcription, and all three Onecut family members as miR-9 targets. Knockdown of ONECUT2 decreases HSV-1 yields in Neuro-2a cells. Overexpression of each ONECUT protein increases HSV-1 replication in Neuro-2a cells, human induced pluripotent stem cell-derived neurons, and primary mouse neurons, and accelerates reactivation from latency in the mouse neurons. Mutagenesis, ChIP-seq, RNA-seq, ChIP-qPCR and ATAC-seq results suggest that ONECUT2 can nonspecifically bind to viral genes via its CUT domain, globally stimulate viral gene transcription, reduce viral heterochromatin and enhance the accessibility of viral chromatin. Thus, neuronal miR-9 promotes viral epigenetic silencing and latency by targeting multiple host transcription factors important for lytic gene activation.

Herpes simplex virus 1 (HSV-1) and HSV-2 are ubiquitous human pathogens within the alphaherpesvirus subfamily. The HSV life cycle switches between productive (lytic) and latent infections. After primary lytic infection in peripheral tissues, HSV undergoes latent infection in neurons of the peripheral nervous system[1]. Reactivation from latency restarts the lytic cycle. The latency-reactivation cycle permits immune evasion and virus spread at times favorable for the virus and is a major obstacle to the prevention and control of herpetic diseases.

HSV lytic and latent infections follow distinct gene expression programs and correspond to different chromatin states of the viral DNA genome[2,3]. During lytic infection, the viral genome enters the nucleus, where it tends to be quickly chromatinized by histones with

---

[1]State Key Laboratory for Diagnosis and Treatment of Infectious Diseases, National Clinical Research Center for Infectious Diseases, Collaborative Innovation Center for Diagnosis and Treatment of Infectious Disease, The First Affiliated Hospital, Zhejiang University School of Medicine, Hangzhou, Zhejiang, China. [2]Department of Medical Microbiology and Parasitology, Zhejiang University School of Medicine, Hangzhou, Zhejiang, China. [3]Zhejiang Provincial Key Laboratory for Microbial Biochemistry and Metabolic Engineering, Hangzhou, Zhejiang, China. [4]Department of Microbiology, Blavatnik Institute, Harvard Medical School, Boston, MA, USA. [5]Department of Biological Chemistry and Molecular Pharmacology, Blavatnik Institute, Harvard Medical School, Boston, MA, USA. [6]Department of Biochemistry and Molecular Biology, Pennsylvania State University, University Park, PA, USA. ✉e-mail: pandongli@zju.edu.cn

repressive heterochromatin modifications. The viral protein VP16 from the entering virion counteracts this repression by complexing with cellular proteins OCT-1 and HCF-1 to recruit activating histone remodelers to the promoters of immediate-early (IE) genes to drive their transcription[4]. Viral IE gene products further stimulate downstream gene transcription. For example, the IE protein ICP0 is an E3 ubiquitin ligase that helps remove heterochromatin modifications and antagonizes other host restrictive mechanisms[5–8] and the IE protein ICP4 binds to viral genes to recruit cellular general transcription factors[9]. Consequently, early (E) and late (L) genes are sequentially expressed leading to production of viral proteins, new viral genomes, and eventually infectious virus particles. During latent infection, the viral genome is highly associated with heterochromatin modifications such as H3K9me3 and H3K27me3[10,11] with only the latency-associated transcript (LAT) locus associated with activating histone modifications[12], correlating with silencing of most viral genes except within that locus[13]. During reactivation from latency, an initial step of widespread de-repression of viral genes mediated by a stress-triggered histone methyl/phospho switch is followed by an ordered cascade of gene expression resembling that of primary infection[14,15].

Besides chromatin-mediated transcriptional regulation, microRNA (miRNA)-mediated post-transcriptional regulation is also implicated in the lytic/latent switch[16]. A cluster of HSV-1 and HSV-2 miRNAs encoded in the *LAT* locus are highly expressed during latency[17–21] and some of them can repress important viral lytic genes, at least in transfection experiments, suggesting this role in latency[22,23]. HSV-1 infection also causes deregulation of cellular miRNAs[24,25], although the role of this deregulation in latency has yet to be determined. In addition, a virus may also exploit host miRNAs specifically expressed in certain host cells to promote viral processes specifically in those cells. HSV appears to undergo latency preferentially in neurons despite its ability to productively infect multiple cell types, implicating a role of neuron-specific host factors in HSV latency. Our previous search for such a role led to the discovery of targeted repression of HSV-1 and HSV-2 ICP0 expression by a neuron-specific host miRNA, miR-138[26,27]. Further investigation revealed that miR-138 also targets the VP16 cofactor OCT-1 and another host transcription factor FOXC1, which stimulates HSV-1 lytic gene expression by decreasing viral heterochromatin, leading to the hypothesis that miR-138 contributes to latency by regulating multiple viral and host genes[28]. However, miR-138 is only one of many neuronal miRNAs. To gain systematic insights into the role of neuronal miRNAs in HSV latency, we performed a functional screen to identify potential regulators of HSV-1 infection followed by investigating the roles of the top miRNA candidate and its targets. In this work, we show that neuronal miR-9 promotes viral epigenetic silencing and latency by targeting multiple host transcription factors that are important for lytic gene activation.

## Results

### A functional screen identified neuron-specific miR-9 as a repressor of HSV-1 neuronal replication

To explore neuron-specific miRNAs regulating HSV-1 infection, we examined the expression profiles of human miRNAs miR-1 to miR-500 one by one in a database (https://ccb-web.cs.uni-saarland.de/tissueatlas/)[29] and found 36 miRNAs with patterns of enriched expression in neuronal tissues. Meanwhile we constructed a virus with the non-essential gC gene of HSV-1 replaced with a luciferase reporter gene (Fig. 1a). This virus, luc-HSV-1 replicates with wild-type (WT) kinetics in African green monkey kidney Vero cells and undifferentiated mouse neuroblastoma Neuro-2a cells (Supplementary Fig. 1a). After transfecting Neuro-2a cells individually with the mimics of the 36 miRNAs and a nonspecific control (NC, sequence shown in Supplementary Fig. 1b), we infected cells with luc-HSV-1 at a multiplicity of infection (MOI) of 0.3, and performed luciferase assays at 72 hours

post-infection (hpi). Relative to the NC, no miRNA significantly increased luciferase activity, but 10 miRNAs, including miR-138, significantly decreased it (Fig. 1b). Notably, the top candidate, miR-9-5p (referred to as miR-9 here), caused a >10-fold decrease in luciferase activity. For the 10 repressive miRNAs, we predicted their targets by TargetScan[30] (https://www.targetscan.org) and combined the top 100 targets for each miRNA for gene ontology (GO) analysis, which showed that the combined putative targets are enriched in molecular functions related to transcriptional activation and the cellular components nucleoplasm, nucleus, and chromatin (Supplementary Fig. 1c), hinting that transcriptional regulation in the nucleus might be a major function of these miRNAs.

No role for miR-9 in herpesvirus infection had been previously reported. miR-9 is conserved in vertebrates and participates in neuronal functions like neurogenesis and axon extension[31–33]. We confirmed its neuron-specific expression by RT-qPCR showing high expression in mouse neuronal tissues including trigeminal ganglia (TG) and dorsal root ganglia (DRG), which are sites of HSV latency, and 100- to 10,000-fold lower expression in non-neuronal tissues (Fig. 1c). In cell culture, miR-9 expression is high in primary neurons isolated from TG and DRG, much lower in Vero and 293T cells (from human embryonic kidney) and extremely low in primary human foreskin fibroblast (HFF) cells. Unexpectedly, the neuroblastoma Neuro-2a cells showed miR-9 levels even lower than Vero and 293T cells, which could be due to aberrant expression in tumor cells. Therefore miR-9 expression is neuron-specific in vivo and in primary cells, but tumor cells may not follow this pattern. Regardless, this observation facilitates tests of the effects of increased miR-9 expression in a neuronal-like environment.

We therefore tested the validity of the screening results by transfecting a miR-9 mimic into Neuro-2a cells followed by HSV-1 infection at a low MOI and observed 5- to 10-fold reductions of virus yields at 48 hpi relative to three different negative control mimics—the NC, a miR-9 seed mutant, and a scrambled miR-9 sequence (Fig. 1d, Supplementary Fig. 1b). We also generated a cell line (N2AmiR9) with doxycycline (Dox)-inducible miR-9 overexpression and a control cell line (N2A-C). In the absence of Dox, there was leaky miR-9 expression from N2AmiR9 cells resulting in miR-9 levels higher than those from N2A-C cells (Supplementary Fig. 2a). Still, Dox treatment substantially increased miR-9 expression from N2AmiR9 but not N2A-C cells. Accordingly, Dox treatment decreased HSV-1 yields in N2AmiR9 but not N2A-C cells (Fig. 1e). Compared to Dox-treated N2A-C cells, HSV-1 infection of Dox-treated N2AmiR9 cells resulted in ~10-fold reduced viral yields, which were reversed by a transfected miR-9 antagomir (Fig. 1f). Experiments conducted in other cell lines showed that transfected miR-9 only slightly reduced HSV-1 yields in HeLa cells and had no significant effect in 293T cells (Supplementary Fig. 2b) raising the possibility that the effects of miR-9 might have required neuronal host factors.

### miR-9 decreases HSV-1 gene expression and increases heterochromatin

To investigate how miR-9 suppressed HSV-1 replication, we analyzed viral gene expression by RT-qPCR. When Neuro-2a cells were transfected with the miR-9 or NC mimic and then infected with HSV-1, the miR-9-treated cells contained significantly reduced levels of *ICP0* (IE gene), *ICP27* (IE gene) and *TK* (E gene) transcripts at 4 hpi (Fig. 1g) suggesting that miR-9 suppresses virus replication by repressing gene expression at or before the IE stage of the lytic cycle. Considering that regulation of gene expression is sometimes related to chromatin remodeling, we performed chromatin immunoprecipitation (ChIP)-qPCR. Relative to Dox-treated N2A-C cells, Dox-treated N2AmiR9 cells infected with HSV-1 for 4 h exhibited elevated association of histone H3 as well as heterochromatin modifications H3K9me3 and H3K27me3 with *ICP0*, *ICP4* and *ICP8* promoters (Fig. 1h, Supplementary Fig. 2c) suggesting that miR-9 represses HSV-1 lytic gene expression by increasing heterochromatin on lytic gene promoters.

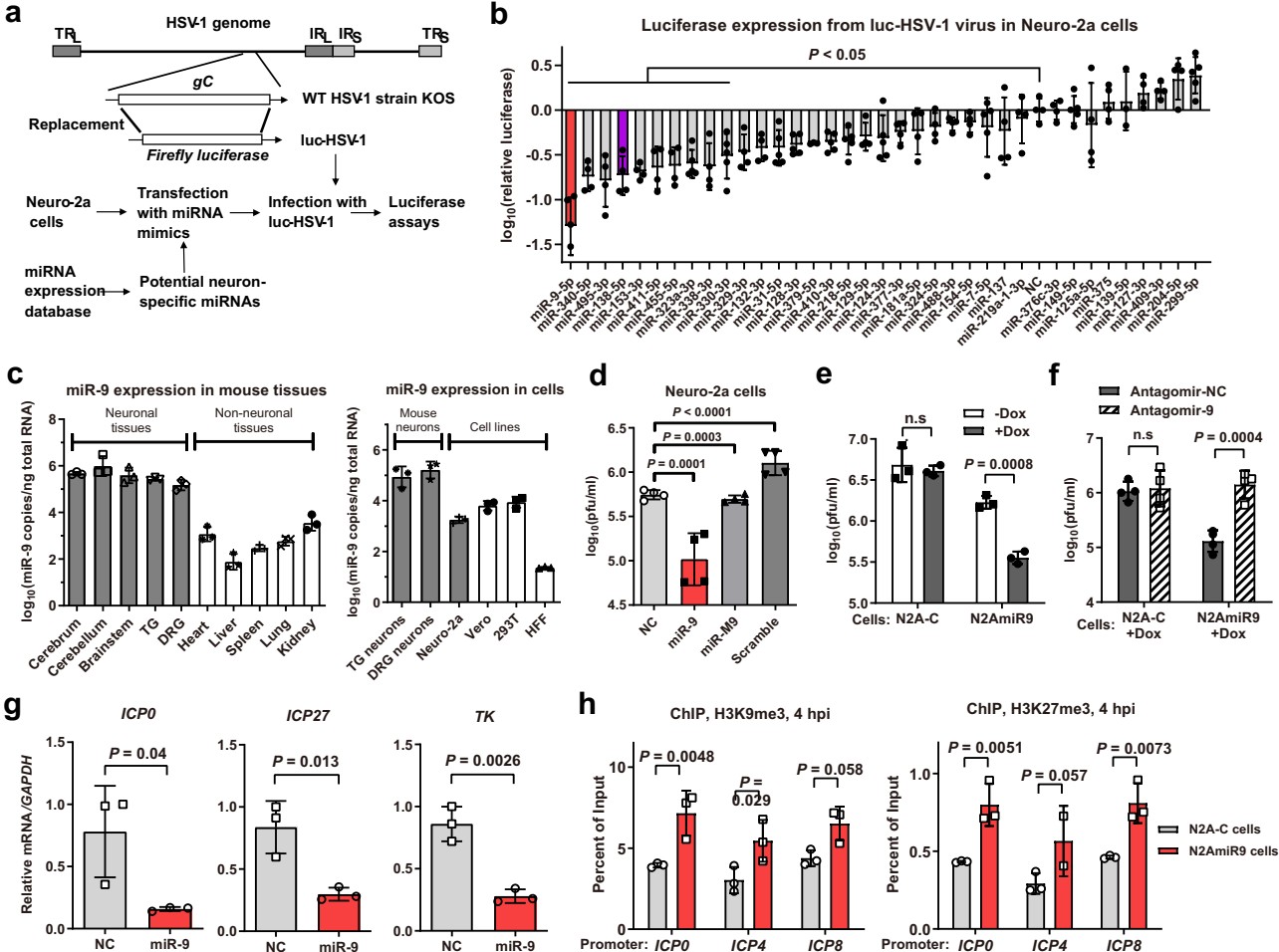

**Fig. 1 | Neuron-specific miR-9 represses HSV-1 replication and gene expression in Neuro-2a cells. a** Diagram of luc-HSV-1 virus and the procedure of screening for neuron-specific miRNAs regulating HSV-1 infection. **b** Neuro-2a cells in 96-well plates were transfected with 20 nM miRNA mimic for 24 h, then infected with luc-HSV-1 (MOI = 0.3) for 72 h before luciferase assays. The adjusted P values: miR-9-5p, <0.0001; miR-340-5p, 0.0002; miR-495-3p, <0.0001; miR-138-5p, 0.0003; miR-153-3p, 0.0007; miR-411-5p, 0.0021; miR-455-5p, 0.0035; miR-323a-3p, 0.0029; miR-338-3p, 0.0030; miR-330-3p, 0.019. **c** RT-qPCR analysis of miR-9 in the indicated tissues (left) or cells (right). **d** Neuro-2a cells were transfected with 20 nM miRNA mimic for 24 h, then infected with KOS (MOI = 0.3) for 48 h before viral titration. **e** The indicated cells were treated or not treated with 1 µg/ml Dox for 48 h, then infected with KOS (MOI = 0.3) for 48 before virus titration. **f** The indicated cells

maintained in 1 µg/ml Dox were transfected with 80 nM antagomir for 24 h, then infected with KOS (MOI = 0.3) for 48 h before viral titration. **g** Neuro-2a cells were transfected with 20 nM miRNA mimic, then infected with KOS (MOI = 1) for 4 h before RT-qPCR analysis of the indicated viral transcripts. **h** N2A-C or N2AmiR9 cells maintained in 1 µg/ml Dox were infected with HSV-1 (MOI = 1) for 4 h before ChIP-qPCR analysis of the enrichment of H3K9me3 (left) or H3K27me3 (right) at the indicated promoters. NC, nonspecific control. $n = 3$ (**c, e, g, h**) or 4 (**b, d, f**) biologically independent samples. Data were analyzed by one-way ANOVA with Dunnett's multiple comparisons tests (**b, d**), two-way ANOVA with Sidak's multiple comparisons tests (**e, f, h**) or two-sided unpaired $t$ tests (**g**) and are presented as the mean ± s.d. Source data are provided as a Source Data file.

## Ectopically expressed miR-9 reduces HSV-1 replication in mouse neurons and TG

To investigate whether miR-9 affects HSV-1 infection of normal neurons, we utilized cultured primary neuron and in vivo models. After isolation and culture of mouse TG neurons, we transduced the neurons with lentivirus to overexpress or inhibit miR-9 before HSV-1 infection. Overexpression was achieved with lentivirus expressing a miR-9 primary transcript (Supplementary Fig. 2d). To inhibit miR-9, we designed "tough decoy" antisense sequences[34]. Among four antisense sequences tested, one (named anti-9, Supplementary Fig. 2e) de-repressed OC2 (a miR-9 target, see below) in Dox-treated N2AmiR9 cells to the greatest extent (Supplementary Fig. 2f). Transduced anti-9 did not affect HSV-1 replication (Fig. 2a). However, transduced miR-9 significantly reduced HSV-1 yields at 24 hpi although the reduction became insignificant at 48 hpi. To facilitate in vivo analysis, we independently generated two miR-9

expressing recombinant viruses (HSV1miR9a and HSV1miR9b) as well as a control virus (HSV1control) (Supplementary Fig. 2g). Ectopic expression of miR-9 from the recombinant viruses was validated by RT-qPCR results showing >2-log increases in miR-9 levels after Neuro-2a cells were infected with HSV1miR9a or HSV1miR9b (Supplementary Fig. 2h). Following inoculating these viruses on mouse cornea, we observed no difference in viral titers in tear film at 1 day post-infection (dpi) (Supplementary Fig. 2i). We then measured viral titers in TG at 7 dpi. Since the data were highly variable, we combined data from four experiments performed under the same conditions to reach enough statistical power. The data showed reduced titers of both miR-9 expressing viruses relative to HSV1control (Fig. 2b). Notably, compared to HSV1control, both HSV1miR9a and HSV1miR9b showed lower fractions of TG containing relative high (>200 pfu) viral titers (Fig. 2c) indicating that miR-9 can attenuate viral replication in mouse TG in vivo.

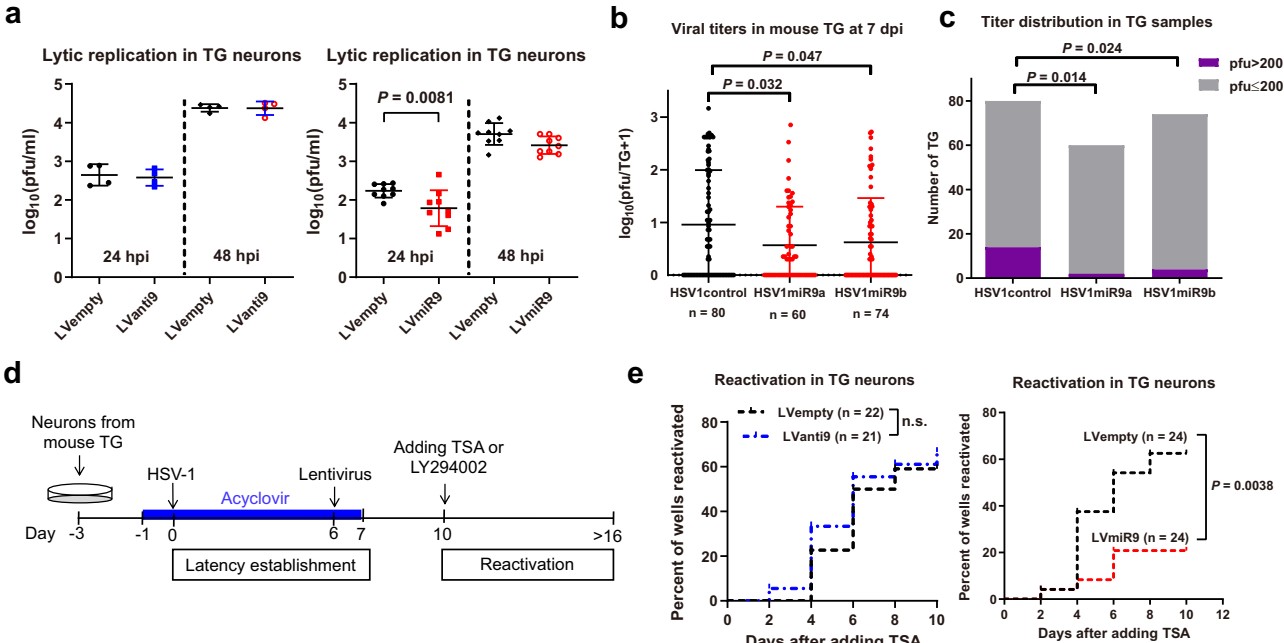

**Fig. 2 | miR-9 suppresses HSV-1 replication and reactivation in mouse TG or TG neurons. a** Neurons isolated from mouse TG were transduced with the indicated lentiviruses and then infected with KOS (MOI = 0.1). At the indicated times, supernatants were collected for virus titration. **b** Mice were inoculated on the cornea with $2 \times 10^5$ pfu/eye. TG were harvested at 7 dpi for viral titration. Each dot represents one TG and the n numbers represent the numbers of TG collected. **c** The numbers of TG with pfu >200 or ≤200 were shown for each virus. **d** Diagram of neuronal culture models of HSV-1 latency and reactivation. **e** Using the model depicted in panel **d**, after the addition of 0.2 μM TSA, supernatants were collected every two days for assays of infectious virus positivity. n.s., non-significant. **a** n = 4 (left) or 9 (right) biologically independent samples. Sample sizes for other panels are labeled on the figure. Data were analyzed by one-way ANOVA with Dunnett's multiple comparisons tests (**b**), two-way ANOVA with Sidak's multiple comparisons tests (**a**), two-sided Fisher's exact tests (**c**) or Log-rank (Mantel-Cox) tests (**e**) and are presented as the mean ± s.d. Source data are provided as a Source Data file.

## miR-9 can repress HSV-1 reactivation from latency in mouse primary neurons

To assess the role of miR-9 in the latency-reactivation cycle, we turned to neuronal culture models, which have proven instrumental for understanding HSV latency[35,36]. We established HSV-1 latency in primary mouse TG neurons by HSV-1 infection in the presence of acyclovir (ACV) (Fig. 2d). A GFP expressing HSV-1, 64-GFP[37], was used to visually monitor the process. After 7 days of ACV treatment, fluorescence signals were reduced to background, at which time we replaced the supernatant with media without ACV. Three days later we added trichostatin A (TSA) or LY294002, each of which had been shown to induce reactivation in neuronal culture models[38,39]. Supernatants were then collected to test for the production of infectious virus. Under our conditions, spontaneous reactivation without any inducer occurred at low rates, and TSA but not LY294002 raised the rates of reactivation significantly above those of spontaneous reactivation (Supplementary Fig. 3). Lentivirus expressing anti-9 showed a trend of increased rate of TSA-triggered reactivation but the increase was not statistically significant (Fig. 2e). The lack of a significant effect of anti-9 during lytic and latent infection might be due to the presence of redundant mechanisms (e.g. miR-138). However, lentivirus expressing miR-9 significantly reduced the rate of TSA-triggered reactivation compared with the control lentivirus. These results suggest that miR-9 can repress HSV-1 replication and reactivation from latency consistent with a role of miR-9 in promoting HSV-1 latency.

## miR-9 targets Oct-1 and Onecut (OC) family genes

miRNAs regulate cellular processes by binding to target mRNAs mainly in their 3' untranslated regions (UTRs) to influence target gene expression[40]. We did not detect cross-regulation between miR-9 and miR-138 (Supplementary Fig. 4a); therefore, we focused on the mRNA targets of miR-9. Alignment of the miR-9 seed sequence to the

3' UTRs of viral transcripts identified only three potential binding sites located in *UL25*, *UL27* and *US7* mRNAs. Since these are L genes, their putative repression by miR-9 is unlikely to explain the early effects of miR-9 on viral gene expression (Fig. 1g). Hence we explored host targets by computational prediction. Interestingly, the VP16 cofactor OCT-1, as well as all three members of ONECUT (OC) transcription factor family (OC1, OC2 and OC3), emerged as the top candidates conserved in vertebrates based on cumulative weighted context score calculated by TargetScan[30]. The 3' UTRs of human and mouse *Oct-1*, *OC2* and *OC3*, and human, but not mouse, *OC1* mRNAs all have multiple sites of the 8mer type (known to be most efficacious miRNA binding site type[40]) (Fig. 3a). Remarkably, there are eight conserved 8mer sites in *OC2* mRNA, two of which (designated sites 7 + 8) overlap with their seed binding sites only 2 nucleotides apart (Supplementary Fig. 4b). There was already previous evidence for miR-9 targeting of human *OC1* and *OC2*[33,41]. Luciferase assays further confirmed that miR-9 represses gene expression through human *Oct-1*, *OC1*, *OC2,* and *OC3* 3' UTRs (Fig. 3b). To provide further evidence that these sites are bound by miR-9, we turned to PAR-CLIP datasets that we previously generated for our miR-138 study (Gene Expression Omnibus Series GSE127503)[28]. Since miR-138 levels were irrelevant for this analysis, we combined data from samples with or without miR-138 overexpression to result in a combined dataset for 293T cells as well as one for Neuro-2a cells. We note that miR-9 was not overexpressed and its endogenous expression is low in these cells so an absence of reads aligned to certain sites does not exclude the possibility of binding in cells where miR-9 is more abundant. Nevertheless, although few reads aligned to the *OC1* or *OC3* mRNA regions possibly reflecting their relative low expression, reads were detected at OC2 sites 3, 4 and 7 + 8, and Oct-1 sites 2 and 4 in 293T cells, and OC2 sites 4 and 7 + 8 and Oct-1 sites 2 and 5 in Neuro-2a cells (Supplementary Fig. 4c). These reads displayed frequent T to C mutations

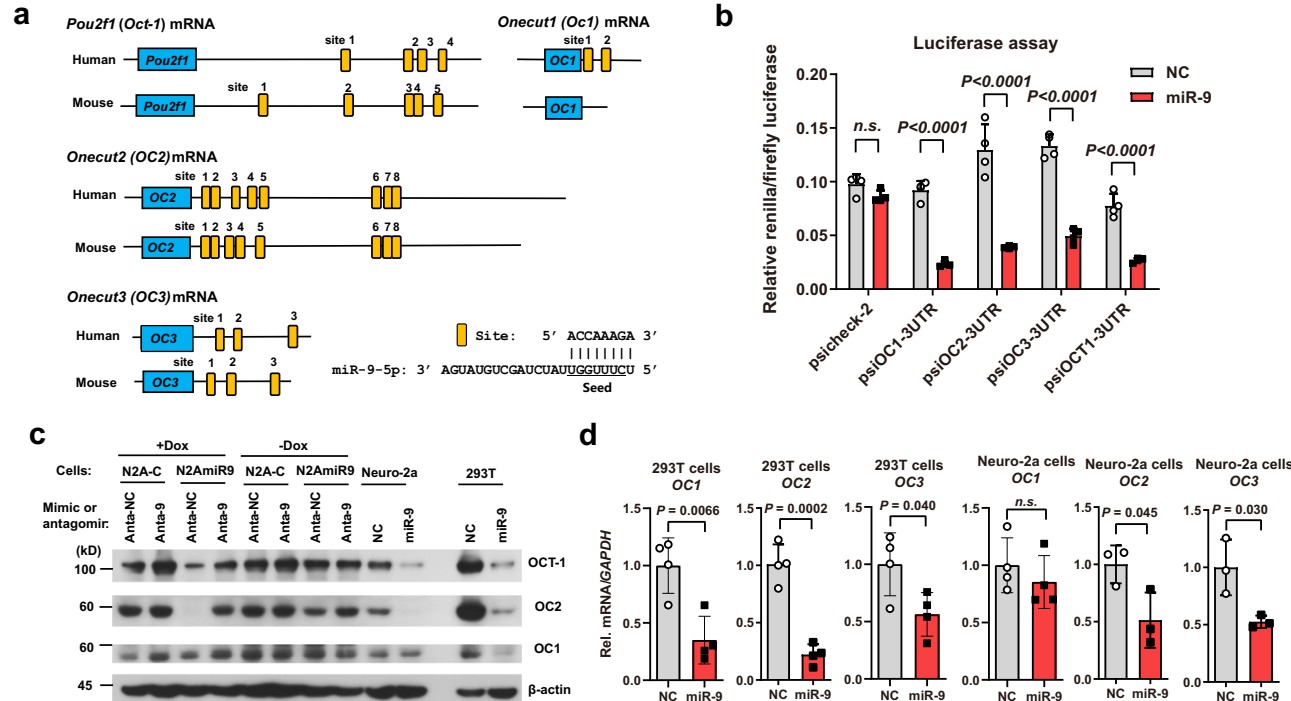

**Fig. 3 | miR-9 targets Oct-1 and OC family transcripts. a** Diagram of miR-9 binding sites in the indicated human and mouse genes. Horizontal lines represent mRNAs. Blue boxes with gene names inside represent open reading frames. Small yellow boxes represent 8mer sites in the 3' UTRs. Base pairing between an 8mer site and miR-9 is shown to the bottom right. **b** 293T cells were co-transfected with 20 nM miRNA mimic and 20 ng/ml plasmid for 48 h before dual luciferase assays. **c** The indicated cells were transfected with 40 nM of the indicated mimics or 80 nM of the indicated antagomirs for 48 h before Western blot analysis of the indicated proteins in the indicated cell lines. This experiment was repeated once with similar results. Anta-NC, non-specific control antagomir. Anta-9, miR-9 antagomir. **d** Neuro-2a or 293T cells were transfected with 80 nM mimic for 48 h before RT-qPCR analysis of human (for 293T cells) or mouse (for Neuro-2a cells) *OC1*, *OC2* or *OC3* mRNA. (**b**, **d**) $n = 4$ biologically independent samples. Data were analyzed by two-way ANOVA with Sidak's multiple comparisons tests (**b**) or two-sided unpaired *t* tests (**d**) and are presented as the mean ± s.d. Source data are provided as a Source Data file.

around the miR-9 seed binding sites characteristic of authentic sites in PAR-CLIP experiments (e.g. Supplementary Fig. 4d). Notably, the overlapping sites 7 + 8 are the strongest *OC2* sites in both cell lines indicating cooperative binding when sites are close to each other.

Next we examined the effects of miR-9 on endogenous expression of Oct-1 and OC transcription factors. Western blot assays revealed that transfected miR-9 strongly repressed OCT-1 and OC2 expression in both 293T and Neuro-2a cells and repressed OC1 expression in 293T but not Neuro-2a cells (Fig. 3c). Moreover, N2AmiR9 cells expressed less OCT-1 and OC2 than N2A-C cells in the presence but not absence of Dox, and, importantly, transfected miR-9 antagomir de-repressed OCT-1 and OC2 in Dox-treated N2AmiR9 cells. As antibodies are not available for all OC proteins, we examined the effects of miR-9 on OC mRNAs using RT-qPCR. Transfected miR-9 decreased *OC1*, *OC2* and *OC3* mRNA levels in 293T cells as well as *OC2* and *OC3* but not *OC1* mRNA levels in Neuro-2a cells (Fig. 3d). These results strongly suggest that miR-9 targets all human and mouse forms of *Oct-1* and *OC* family mRNAs except mouse *OC1* which lacks miR-9 sites.

## OCT-1 and OC2 are important for efficient HSV-1 replication in neuronal cells

OCT-1 is known to be important for IE gene transcription[42]. Consistently, two independent OCT-1 knockout cell lines both exhibited reduced HSV-1 yields relative to the parental Neuro-2a cells (Fig. 4a, Supplementary Fig. 5a). For the OC proteins, we first estimated their endogenous expression using RT-qPCR and found that in both Neuro-2a cells and mouse TG neurons, *OC2* expression is higher than *Oct-1*, which is higher than *OC1* and *OC3* (Supplementary Fig. 5b). Therefore, we attempted to knock out OC2 from Neuro-2a cells. However, multiple attempts using our standard CRISPR/Cas9 protocol failed, hinting that OC2 might be important for cellular survival or function. We therefore transiently knocked down OC2 and found that each of two siRNAs capable of depleting OC2 significantly decreased HSV-1 yields in Neuro-2a cells (Fig. 4b, Supplementary Fig. 5a). Therefore, OCT-1 and OC2 are important for efficient HSV-1 replication in neuronal cells and their repression by miR-9 should contribute to miR-9 suppression of HSV-1 replication.

## OC proteins can promote HSV-1 replication and reactivation from latency

Since OC proteins had not been reported to regulate herpesviruses, we further investigated these proteins. Consistent with the results of the OC2 knockdown experiments, overexpression of any of the three human OC proteins as well as mouse OC2 by plasmid transfection significantly increased HSV-1 yields in Neuro-2a cells (Fig. 4c, Supplementary Fig. 5c–d). Lentiviral transduction of each human OC gene into HFF cells also considerably increased HSV-1 yields (Fig. 4d). However, transfection of these genes had little effect on virus yields in 293T cells (Supplementary Fig. 5e). Therefore, the functions of the OC proteins in HSV-1 infection are conserved between mice and humans but might be cell-type specific. Interestingly, in both Neuro-2a and HFF cells, the effects of OC2 were more dramatic on ICP0-null virus than WT virus (Fig. 4c, Supplementary Fig. 5f) indicating that OC proteins might compensate for certain ICP0 functions.

We then further assessed the effects of human OC proteins in primary neuronal culture. When we transduced mouse TG neurons with OC2 expressing or control lentivirus followed by infection with the GFP expressing HSV-1, the increase in visualized GFP expression caused by OC2 was obvious (Fig. 4e). Then we infected such neurons with WT HSV-1 strain KOS after lentivirus transduction and observed that each transduced OC gene could strongly promote HSV-1 replication with ~10-

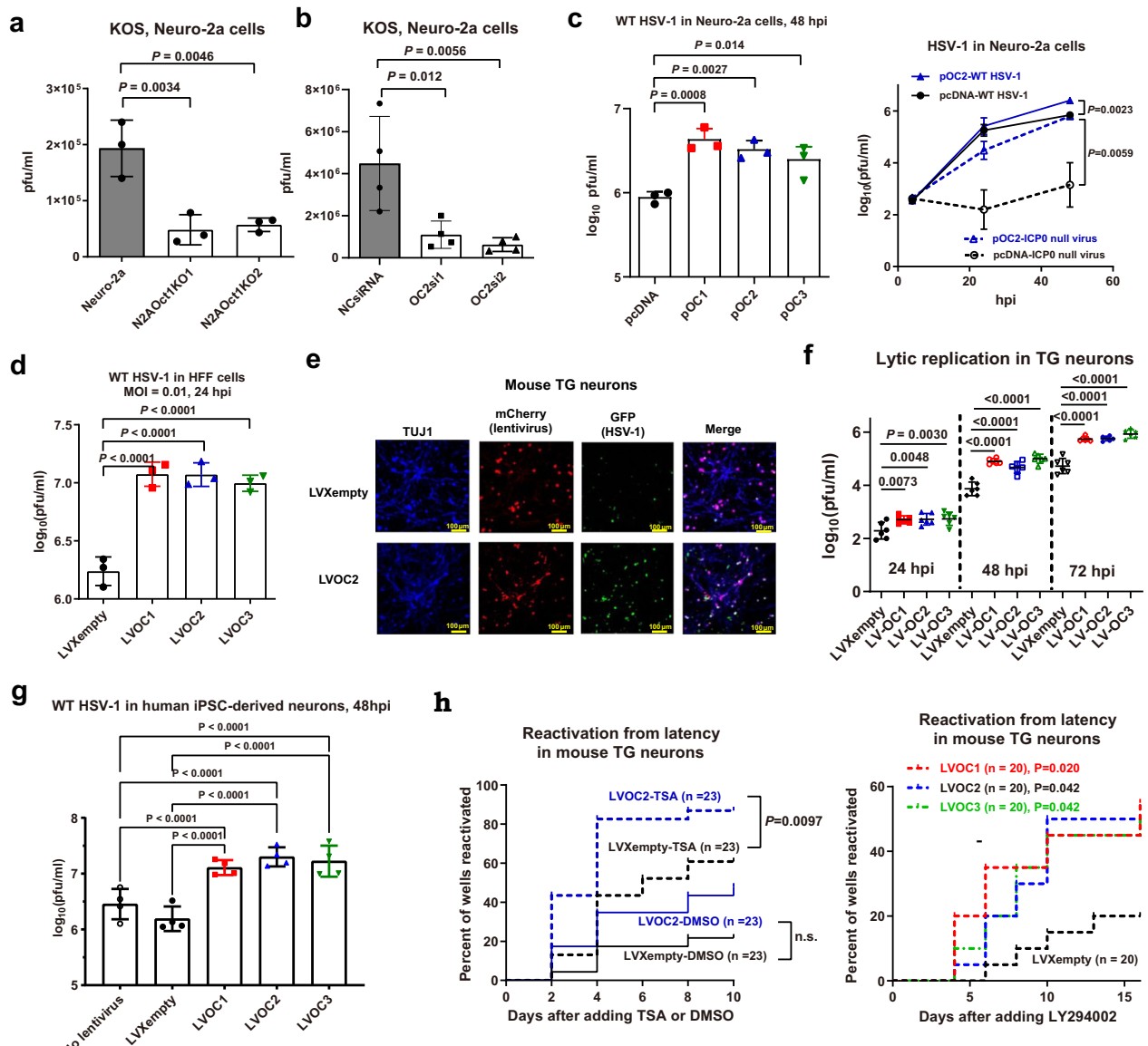

**Fig. 4 | OCT-1 and OC proteins promote HSV-1 replication and/or reactivation.**
**a** The indicated cells were infected with KOS (MOI = 0.3) for 48 h before virus titration. **b** Neuro-2a cells were transfected with 80 nM siRNA for 40 h, then infected with KOS (MOI = 0.2) for 48 h before virus titration. **c** Neuro-2a cells were transfected with 200 ng/ml plasmid for 24 h, then infected with WT KOS or ICP0-null virus (MOI = 0.2) for 48 h (left) or the indicated times (right) before virus titration. **d** HFF cells were transduced with lentivirus for 72 h, then infected with KOS (MOI = 0.05) for 48 h before virus titration. **e** Primary TG neurons were transduced with lentivirus for 4 days, then infected with 64-GFP (MOI = 0.1) for 48 h before immunofluorescence analysis of TUJ1 and visualization of GFP (green) and mCherry (red) expression. The experiment was repeated once with similar results. **f** Primary TG neurons were transduced with lentivirus for 4 days, then infected with KOS (MOI = 0.1) for the indicated times before titration of the virus in the

supernatants. **g** Differentiated human neurons were transduced with lentivirus for 10 days, then infected with KOS (MOI = 1) for 48 hpi before virus titration. The data are from 4 biologically independent experiments with 4 replicates per experiment. Each point represents the mean from one experiment. **h** TG neurons were transduced with lentivirus and supernatants were assayed for infectious virus positivity at the indicated days after addition of DMSO (left), TSA (0.2 μM) (left) or LY294002 (20 μM) (right). $n = 3$ (**a**, **c**, **d**), 4 (**b**), 6 (**f**) or 16 (**g**) biologically independent samples. Sample sizes for (**h**) are labeled on the figure. Data were analyzed by one-way ANOVA with Dunnett's multiple comparisons tests (**a**, **b**, **d**, **f** and left panel of **c**), two-sided unpaired $t$ tests (right panel of **c**), two-way ANOVA, matched by experimental batch, with Šídák's multiple comparisons tests (**g**) or Log-rank (Mantel-Cox) tests (**h**) and are presented as the mean ± s.d. Source data are provided as a Source Data file.

fold increases in HSV-1 yields at 48 and 72 hpi (Fig. 4f). To test this further in human neurons, we derived neurons from human induced pluripotent stem cells (iPSCs), transduced the neurons with lentivirus before KOS infection and again found that each transduced OC gene increased HSV-1 yields by ~10-fold at 48 hpi (Fig. 4g). Lastly, we tested the effects on reactivation from latency using the latency-reactivation model in mouse TG neurons (Fig. 2d). While in the absence of a chemical trigger transduced OC2 caused a trend of more frequent reactivation without statistical significance, in the presence of TSA it significantly

increased the rate of reactivation (Fig. 4h). Moreover, transduction of any of the three OC genes significantly increased the rate of reactivation in the presence of LY294002. These results suggest that each OC protein can facilitate HSV-1 replication and reactivation from latency in neurons.

## OC proteins and miR-9 regulates HSV-2 neuronal replication
These effects on HSV-1 made us wonder whether OC proteins regulate HSV-2 infection. Indeed, knockdown of OC2 significantly reduced HSV-2 yields (Supplementary Fig. 6a) and all three OC

proteins could significantly increase HSV-2 yields when over-expressed in Neuro-2a cells (Supplementary Fig. 6b). Given these results, we tested whether the repressor of these OC proteins, miR-9, had opposite effects. Indeed, transfection of the miR-9 mimic and stable overexpression of miR-9 in Neuro-2a cells both significantly reduced HSV-2 yields compared to their respective controls (Supplementary Fig. 6c–d).

### OC2 stimulation of HSV-1 replication requires CUT domain binding to DNA backbone

All OC proteins have two highly conserved DNA binding domains (DBDs) including a CUT domain and a homeodomain (HOX domain) connected by a short linker, as well as upstream undefined regions collectively designated the N-terminal domain

(NTD) (Fig. 5a). Transfection-infection experiments in Neuro-2a cells using plasmids expressing truncated mutants of OC2 and ICP0-null virus showed that the CUT domain and NTD are both required for stimulation of HSV-1 replication (Fig. 5b, Supplementary Fig. 5c). Curiously, the HOX domain contributes negatively as its removal made the protein a stronger stimulator. Because of this, the following experiments used OC2ΔHOX to dissect the activating mechanisms. Within the NTD, a region of aa113-217 contributes significantly to stimulation. We then mutated the CUT domain at residues contacting DNA based on a crystal structure of the bipartite OC1 DBDs complexed with cognate DNA[43]. Interestingly, the residues predicted to contact bases were all dispensable for stimulation whereas each of the two residues predicted to contact the DNA backbone, glutamine (Q)

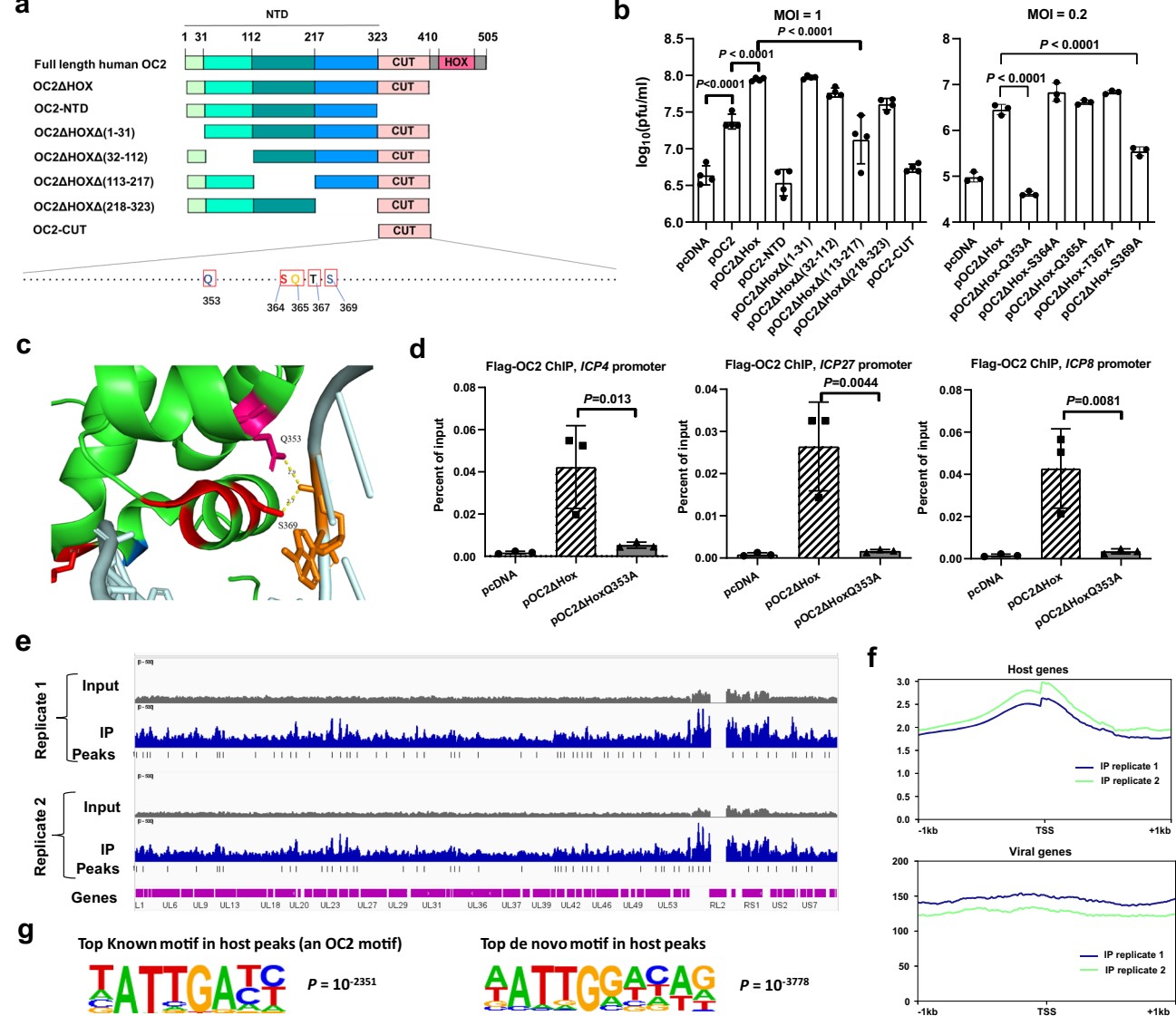

**Fig. 5 | OC2 binds to HSV-1 DNA to stimulate viral gene transcription.**
**a** Schematic of OC2 and its truncated mutants. Amino acid positions are labeled at the top. Expanded below are CUT domain residues with the mutated ones in boxes. **b** Neuro-2a cells were transfected with 200 ng/ml plasmid for 24 h, then infected with ICP0-null virus for 48 h before virus titration. **c** Crystal structure of OC1 DBDs complexed with DNA adapted by using the PyMOL software (PDB code 2D5V). The dotted lines represent hydrogen bonds whose distances are labeled. **d** Neuro-2a cells were transfected with 400 ng/µl plasmid for 40 h, then infected with ICP0-null virus (MOI = 3) for 5 h before ChIP-qPCR analysis of the enrichment of transfected Flag-tagged proteins at the indicated promoters using a Flag antibody. **e** After ChIP

performed as in panel **d**, DNA samples were sequenced. In gray and blue are coverage plots along the HSV-1 genome (with the terminal repeat sequences deleted) for input and immunoprecipitated DNA samples, respectively. Small vertical sticks represent identified peaks in the immunoprecipitated samples. **f** ChIP-seq signal around transcription start sites (TSS) for host (upper) or viral (lower) genes. **g** The top-ranked known (left) or de novo (right) motifs identified in host peaks with the *P* values displayed. *n* = 2 (**e**–**g**), 3 (**d**, right panel of **b**) or 4 (left panel of **b**) biologically independent samples. Data were analyzed by one-way ANOVA with Dunnett's multiple comparisons tests (**b**, **d**) and are presented as the mean ± s.d. Source data are provided as a Source Data file.

353 and serine (S) 369, was essential. In the crystal structure, the residues corresponding to Q353 and S369 both make hydrogen bonds with a phosphodiester oxygen atom on the DNA backbone (Fig. 5c).

## Widespread binding to sites on the HSV-1 genome by OC2

The requirement for a DBD led to the question of whether OC2 binds to viral DNA. Searching for a known consensus sequence of the OC family (ATCRAT) (https://jaspar.genereg.net/) identified only one site within the HSV-1 genome (in the L gene *UL38*). To explore the possibility of binding independent of this consensus, we performed ChIP-qPCR by pulling down OC2ΔHOX with a Flag antibody after Flag-tagged-OC2ΔHOX transfected Neuro-2a cells were infected with ICP0-null virus. Relative to the empty vector, OC2ΔHOX was significantly enriched on *ICP4, ICP27,* and *ICP8* promoters at both 2 and 5 hpi (Fig. 5d, Supplementary Fig. 7). Substitution of the Q353 residue largely disrupted the enrichment indicating that efficient OC2 binding to viral DNA requires this residue. To more systematically investigate the binding pattern, we conducted ChIP-seq on OC2ΔHOX transfected, ICP0-null virus infected Neuro-2a cells harvested at 5 hpi. Using input DNA samples as controls, OC2ΔHOX displayed widespread binding to the HSV-1 genome with numerous peaks identified that were consistent between two biological replicates (Fig. 5e). Unlike the ChIP-seq signals on the host genome, the signals on the viral genome showed no enrichment at transcription start sites (TSS) (Fig. 5f). When we performed motif search in host peaks, the top-ranked known motif was indeed a previously identified OC2 binding motif (containing ATTGAT)[44] (Fig. 5g) suggesting that OC2ΔHOX specifically binds to canonical OC2 sites in the host genome. When we ran a de novo motif search for the host peaks, a slightly modified motif containing ATTGG(A/T) was identified. However, no motif was identified in the viral peaks and only a small fraction of the viral peaks contain the ATTGG(A/T) motif suggesting that OC2ΔHOX binds to the HSV-1 genomes with little sequence specificity.

## OC2 can globally stimulate the initial stage of HSV-1 gene transcription

Since OC proteins are transcription factors binding to viral DNA, we hypothesized that they regulate viral gene transcription. Indeed, transfected OC2ΔHOX substantially increased the mRNA levels of the IE gene *ICP27*, E gene *TK* and L gene *gC* at 3 and 6 h after Neuro-2a cells were infected ICP0-null HSV-1 (Fig. 6a). These effects were observed regardless of the presence of ACV, and viral DNA levels were unaffected by OC2ΔHOX at 3 hpi (Fig. 6b) suggesting that the impact is independent of DNA replication. We also excluded regulation of steps upstream of transcription by showing no effect on viral attachment to the cell or entry of viral DNA into the nucleus (Supplementary Fig. 8). To more systematically assess the impact, we performed RNA-seq and observed upregulation of all viral transcripts, regardless of IE, E or L classification, at 4 hpi in OC2ΔHOX transfected cells with fold changes ranging from 2 to 10 (Fig. 6c). Therefore, OC2 globally stimulates viral gene transcription starting very early in infection.

## OC2 can decrease heterochromatin and increase the accessibility of HSV-1 chromatin

To understand the early impact on viral gene transcription and the widespread binding of OC2 to viral DNA, we tested the possibility that OC2 might interact with viral proteins VP16 and ICP4, which are important for the initial stage of viral gene transcription and associate with the viral genome. However, co-immunoprecipitation experiments showed no evidence that OC2 interacts with VP16 or ICP4 expressed from KOS at 12 hpi (Supplementary Fig. 9). To test whether the effects on viral gene transcription were related to chromatin remodeling, we performed ChIP-qPCR to quantify the enrichment of total histone H3, heterochromatin marks H3K9me3 and H3K27me3, as well as a mark of active chromatin H3K27ac. At 2 h after ICP0-null virus infection of Neuro-2a cells, transfected OC2ΔHOX significantly decreased H3K9me3 on ICP4, ICP27 and ICP8 promoters, decreased total H3 on ICP4 and ICP27, but not ICP8, promoters, but had no significant effect

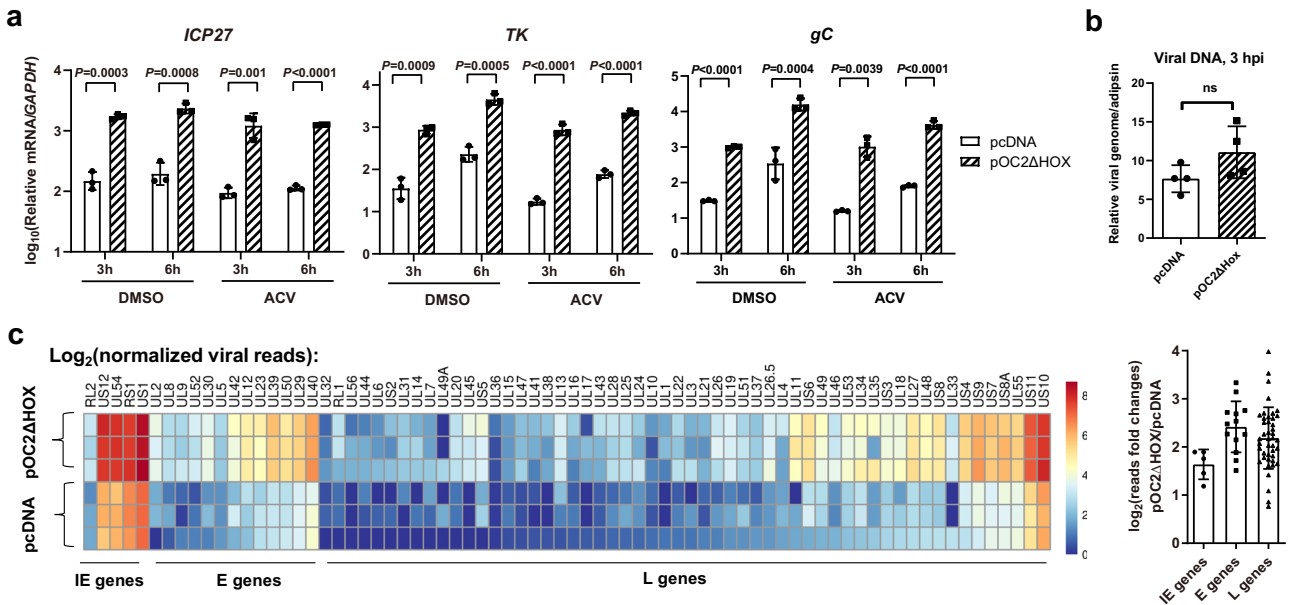

**Fig. 6 | OC2 can globally stimulate HSV-1 gene transcription. a** Neuro-2a cells were transfected with 200 ng/ml plasmid for 42 h, then infected with ICP0-null virus (MOI = 1) in media containing 100 μM ACV or DMSO for 3 or 6 h before RT-qPCR analysis. **b** Neuro-2a cells were transfected with 200 ng/ml plasmid, then infected with ICP0-null virus (MOI = 1) for 3 h before viral DNA quantification. **c** Neuro-2a cells were transfected with 200 ng/ml plasmid for 42 h, then infected with ICP0-null virus (MOI = 2) for 4 h before RNA-seq analysis. Normalized reads for each viral transcripts were log2 transformed before illustration by the Heatmap. The right panel shows the fold changes for genes of different kinetic classes with each point representing a viral transcript. $n = 3$ (**a, c**) or 4 (**b**) biologically independent samples. Data were analyzed by two-way ANOVA with Sidak's multiple comparisons tests (**a**) or two-sided unpaired $t$ tests (**b**) and are presented as the mean ± s.d. Source data are provided as a Source Data file.

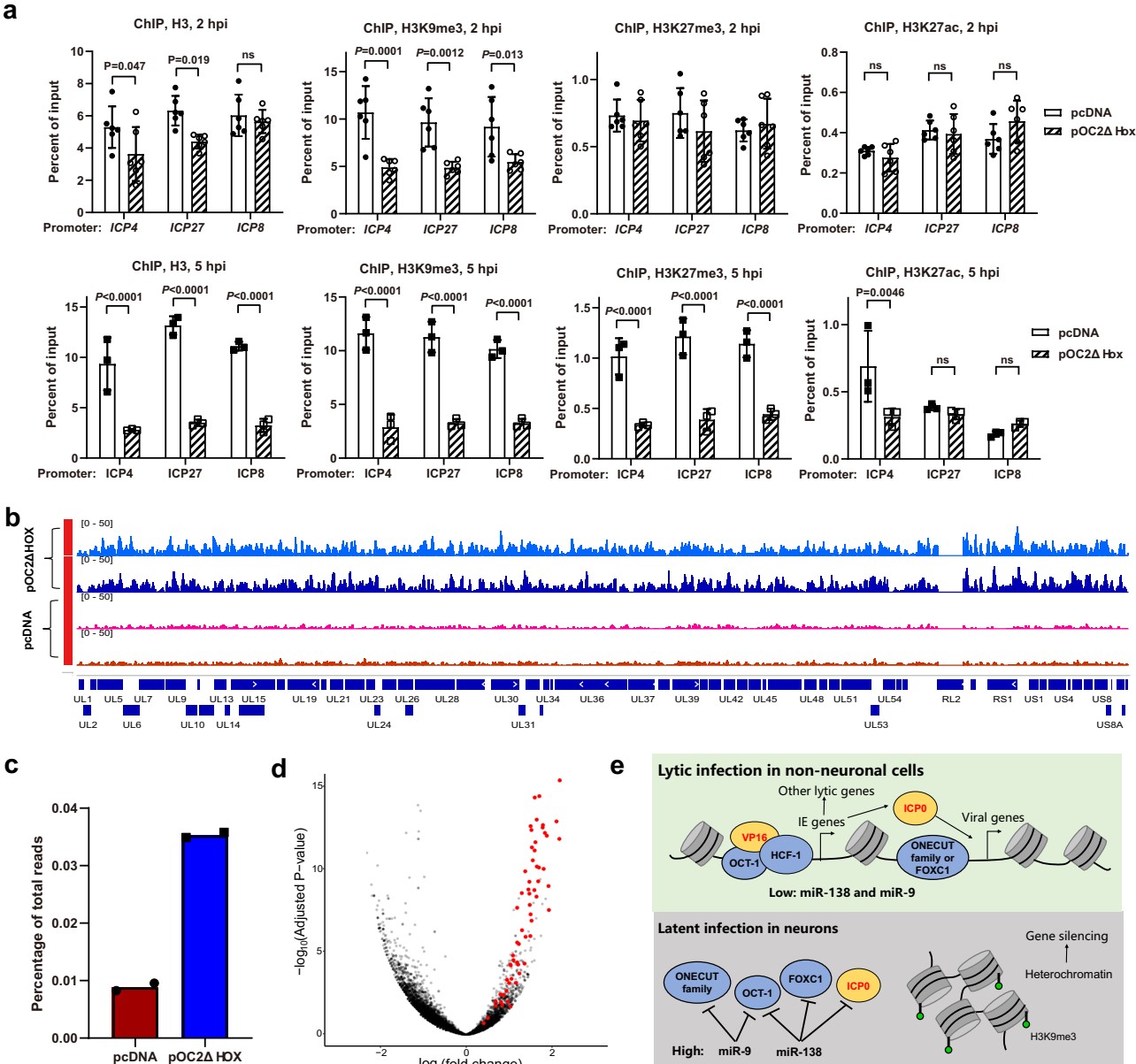

**Fig. 7 | OC2 can regulate HSV-1 chromatin. a** Neuro-2a cells were transfected with 600 ng/µl plasmid for 43 h, then infected with ICP0-null virus (MOI = 2) for 2 or 5 h before ChIP-qPCR analysis of the enrichment of histone H3 and its modifications on the indicated promoters. **b** Neuro-2a cells were transfected with 200 ng/ul of pcDNA or pOC2ΔHOX for 40 h, then infected with ICP0-null virus (MOI = 1) for 2 h before ATAC-seq analysis. Coverage plots for reads aligned to the entire ICP0-null virus genome are shown. **c** Percentages of total viral reads within total reads in the ATAC-seq data. **d** Volcano plot for the differences between pOC2ΔHOX and pcDNA transfected cells in reads aligned to viral (red) or host (black) promoters. Each dot represents one gene promoter. **e** Model of regulation of the lytic/latent balance by host neuronal miRNAs and their targets. n = 2 (**b**, **c**, **d**), 3 (lower panels of **a**) or 6 (upper panels of **a**) biologically independent samples. Data were analyzed by two-way ANOVA with Sidak's multiple comparisons tests (**a**), two-sided unpaired t tests (**c**) or two-sided Fisher's t tests adjusted by false discovery rates to correct for multiple testing (**d**). The data are presented as the mean ± s.d. Source data are provided as a Source Data file.

on H3K27me3 or H3K27ac on any of these promoters (Fig. 7a). At 5 hpi, OC2ΔHOX substantially decreased the association of histone H3, as well as H3K9me3 and H3K27me3 modifications with all three promoters while having little effect on H3K27ac association. When normalized to total H3, H3K9me3 and H3K27me3 were largely unaffected but H3K27ac per H3 on ICP27 and ICP8 promoters was increased (Supplementary Fig. 10a) suggesting that OC2ΔHOX causes removal of repressive chromatin but retains activating chromatin. Another experiment analyzing cells harvested at 4 hpi confirmed the effects on H3K9me3 and additionally showed that the effects depended on the Q353 residue indicating that binding to viral DNA is important for heterochromatin removal (Supplementary Fig. 10b).

To further assess the impact of OC2 on viral chromatin, we performed transposase-accessible chromatin with high-throughput sequencing (ATAC-seq) after plasmid transfection and ICP0-null virus infection of Neuro-2a cells. Notably, at the very early time of 2 hpi, OC2ΔHOX caused substantially increased viral reads associated with accessible chromatin across the entire viral genome (Fig. 7b) resulting in ~4-fold increased fractions of viral reads within total reads (Fig. 7c). Two biological replicates showed similar results. When we plotted the reads differences for each viral or host gene promoter in a volcano plot, all viral promoters showed increased accessibility following OC2ΔHOX transfection whereas host promoters showed a wide range of differences with no net global effect (Fig. 7d). Therefore, as

OC2 globally increases viral gene transcription, it also globally increases the accessibility of viral chromatin.

## Discussion

Silencing of lytic genes and heterochromatinization of the viral genome are hallmarks of HSV-1 latency, yet the mechanisms involved remain obscure. The viral *LAT* locus has been shown to contribute to these processes[10,45] but there is growing evidence that host factors are important (e.g[39,46–48]). We propose a model in which neuronal miRNAs including miR-9 and miR-138 contribute to a cellular environment favorable for HSV-1 latency. These miRNAs form a network of interactions with their targets (Fig. 7e). A general function of these targets, including OC proteins, OCT-1, FOXC1 and ICP0, during HSV-1 infection is stimulation of early events of HSV-1 lytic gene expression. In non-neuronal cells where expression of miR-9 and miR-138 is low, relatively high expression of these target proteins would enable quick onset of the lytic cycle. However, in neurons, the expression of each of these proteins is strongly repressed by at least one of the two miRNAs (OCT-1 is repressed by both) through multiple miRNA binding sites[26,28]. We note that these genes should be also under transcriptional control but their simultaneous repression by the miRNAs should prevent leaky high expression of any of these genes and the collective effect should strongly favor latency. Moreover, our screen also identified other neuronal miRNAs that potentially repress HSV-1 replication raising the possibility that they also contribute to HSV-1 latency. Interestingly, the neuronal miRNAs have an enriched function in targeting transcription factors (Supplementary Fig. 1c). The miRNA-transcription factor network, which has yet to be thoroughly explored, might be essential for neuronal physiology, and HSV appears to exploit the host network to help achieve persistence in neurons.

In the nervous system OC proteins play a role mainly in neuronal development[49,50]. For example, in mouse embryos, absence of one or more OC proteins resulted in defects in the trigeminal central projections and loss of neurons in the rhombencephalic mesencephalic trigeminal nucleus[51,52]. However the only previous report on OC protein regulation of virus infection of which we are aware was about inhibition of hepatitis B virus replication by OC1 in hepatocytes, which involved a CUT-domain independent mechanism[53]. Here we reveal unique mechanisms of regulation of HSV-1 infection by OC proteins. While we focused on OC2 for mechanistic investigation, we assume that similar mechanisms are employed by OC1 and OC3 given their sequence conservation. At least for OC2, regulation of HSV-1 infection appears to be initiated by binding of the CUT domain to viral DNA since at very early times OC2ΔHOX binds to viral DNA and causes viral chromatin perturbation and the Q353 residue within the CUT domain that is important for viral DNA binding is essential for effects on H3K9me3 deposition and virus replication. On the other hand, the HOX domain is dispensable for these events consistent with a previous report that the HOX domain is not always required for DNA binding by OC1[54] although the published crystal structure shows that both DBDs of OC1 interact with multiple atoms of both DNA bases and the backbone[43]. Our ChIP-qPCR and ATAC-seq data suggest that following viral DNA binding, OC2 remodels HSV-1 chromatin to result in more accessible chromatin conducive to active transcription. While this is consistent with a report that OC proteins can remodel cellular chromatin accessibility[50], we show that OC2ΔHOX globally increased the accessibility of viral promoters whereas it had both positive and negative effects on the accessibility of host promoters with no net effect, indicating differential regulation of viral and host chromatins by OC proteins. The global impact on viral gene expression and chromatin is consistent with widespread binding to viral DNA with little sequence specificity. Like other sequence-specific DNA binding proteins, OC proteins doubtless have some weaker, basal nonspecific DNA binding activity, which we speculate would depend on hydrogen bonding between the CUT domain and DNA backbone (Fig. 5c). HSV-1

chromatin, at least during lytic infection, is known to be less compacted than host chromatin[55]. The relatively open viral chromatin might allow OC proteins to bind in a nonspecific and widespread manner to further increase chromatin accessibility. In contrast, the largely compacted host chromatin permits OC proteins access to only certain genomic regions and stable binding to such regions might require sequence specific interactions. Since the NTD is also required for virus stimulation, we hypothesize that while the CUT domain binds to the viral genome, the NTD interacts with other proteins, although evidently not HSV-1 ICP4 or VP16, to destabilize viral heterochromatin, resulting in enhanced chromatin accessibility. More work is required to elucidate exactly how OC proteins remodel viral chromatin to tilt the lytic/latent balance.

In addition to their roles in HSV-1 infection and latency, we provide evidence that miR-9 and the OC proteins can also regulate HSV-2 neuronal replication. We have previously shown that the miR-138-*ICP0* interaction is functionally conserved between HSV-1 and HSV-2, and host OCT-1 and FOXC1 are important for HSV-2 neuronal replication too[27]. Furthermore, given the sequence nonspecific effects of the OC proteins on viral chromatin, these proteins might regulate infection by other viruses whose genomes form chromatin in the nucleus, especially those in the alphaherpesvirus subfamily, which generally infect neurons. Therefore, the possibility of the existence of a conserved neuronal network controlling infection and latency by different alphaherpesviruses deserves consideration. Future studies are needed to further explore the other components of the network and their roles during infection by HSV and other alphaherpesviruses.

## Methods
### Cells

Vero (catalogue no. CCL-81), 293T (catalogue no. CRL-3216), U2OS (catalogue no. HTB-96), HFF (catalogue no. CRL-1634), HeLa (catalogue no. CCL-2) and Neuro-2a (catalogue no. CCL-131) cells were obtained from American Type Culture Collection and maintained as described previously[56,57]. N2AOct1KO cell lines was described previously[27]. Construction of N2AmiR9 and N2A-C is described below.

TG neurons were isolated and cultured as described previously with slight modifications[58]. Briefly, 6-week-old Institute for Cancer Research male mice (Shanghai Laboratory Animals Center) were anesthetized with isoflurane (RWD) for 1 min and transcardially perfused with approximately 10 mL of phosphate-buffered saline (PBS). TG neurons were dissected and digested in collagenase/dispase II solution (Sigma-Aldrich, C9891 and D4693) at 37 °C for 1 h. Neurons were purified by gradient separation in an OptiPrep gradient (Sigma-Aldrich, D1556) followed by two washes with Neurobasal-A medium (Thermo Fisher Scientific, 10888022) with 2% N21 (Bio-Techne, AR008). Purified neurons were counted and plated on 10-mm coverslips pretreated with poly-D-lysine (Sangon Biotech, E607014-0002) and laminin (Sigma-Aldrich, L2020). About 5000 neurons were plated on each coverslip. After the neurons adhered, the coverslips were transferred to a 48-well plate and cultured in Neurobasal-A medium with 2% N21 (Bio-Techne, AR008), 50 ng/mL β-neuronal growth factor (Novoprotein, C060), 50 ng/mL glialderived neurotrophic factor (Novoprotein, C226) and 1 mg/mL mitomycin C (MedChemExpress, HY-13316).

Neurogenin3 inducible human iPSCs (iNGN3 iPSCs) (Ng et al.[59]) were kindly provided by George Church of Harvard Medical School and differentiated as described previously with some modifications[28]. In addition to Neurogenin3 induction, the iNGN3 iPSCs were treated with small molecules (LDN193189, SB-431542, CHIR99021, DAPT, and SU5402) to enhance neuronal differentiation as described previously[60,61].

### Viruses

HSV-1 strain KOS, ICP0-null mutant 7134 virus, WT-BAC virus, and HSV-2 strain 186 were propagated and assayed as described previously[27,28].

64-GFP virus with an insertion of the GFP gene between UL49 and UL50 of KOS was kindly provided by David Leib (Geisel School of Medicine at Dartmouth)[37]. Virus propagation and titration by plaque assays were described previously[19]. Luc-HSV-1 was constructed by using two-step Red-mediated bacterial artificial chromosome (BAC) recombination as previously described[62]. The luciferase reporter gene (*luc2*) was isolated from the pGL4.13 plasmid (Promega) using HindIII and BamHI enzymes and cloned into the pSP72 vector. An excisable kanamycin resistance marker I-SceI-*aphAI* was PCR amplified from a plasmid PLAY2-I-SceIAphAI (kindly provided by J. Kamil of Louisiana State University Medical Center) using primers AgeLucKan Fw and Age-LucKan Rv and cloned into the AgeI site of pSP72-luc2 plasmid. This plasmid was then used as a PCR template to generate the transfer construct to replace the HSV-1 gC open reading frame using gCLuc2-FOR and gCLuc2-REV primers. The PCR product was gel purified and electroporated into GS1783 *E. coli* cells harboring the BAC of the WT-BAC virus. The resulting BAC was subjected to a second step of recombination to remove the kanamycin cassette. The integrity of BACs was verified by electrophoresis of HindIII digested products. The final BAC DNA was transfected into Vero cells to generate HSV1-luc virus. A similar method was used to construct HSV-1control, HSV-1miR9a and HSV-1miR9b. Briefly, pcDNA3.1-kana-ICP47-miR-9 plasmid with the kanamycin-resistance gene, I-SceI site, pri-miR-9-2 and homologous sequences inserted into FLAG-HA-pcDNA3.1 was synthesized by Tsingke Biotechnology. The synthetic DNA sequences are listed in Supplementary Table 1. This plasmid was used as a template with miR-TransformF and miR-TransformR primers to PCR amplify the DNA cassettes used for recombination according to the two-step red-mediated recombination protocol. Primer sequences are listed in Supplementary Table 2.

## Plasmids

pOC1(human), pOC2(human), pOC2(mouse) and pOC3(human) plasmids were obtained from MiaoLingBio. To construct psiOC1-3UTR, psiOC2-3UTR, psiOC3-3UTR and psiOCT1-3UTR, human OC1, OC2, OC3, and Oct-1 3′ UTR fragments were amplified by PCR using 293T cell suspension as a template and OC1-3UTR-F and OC1-3UTR-R, OC2-3UTR-F and OC2-3UTR-R, OC3-3UTR-F and OC3-3UTR-R primers, respectively, and inserted between the XhoI and NotI sites of psiCheck-2 (Promega). To construct pTRIPZmiR9, pri-miR-9-2 sequence was amplified by PCR using the pcDNA3.1-kana-ICP47-miR-9 template and pTRIPZ-miR-9-F and pTRIPZ-miR-9-R primers, and inserted between XhoI and MluI sites of pTRIPZ (Thermo Scientific). The plasmids expressing human OC2 mutants were constructed using a NEB Q5 Site-Directed Mutagenesis Kit. For example, PCR was performed using the pOC2(human) template and OC2ΔHOX-F and OC2ΔHOX-R primers to generate linear pOC2Δ-HOX. Then the PCR products were incubated with a Kinase-Ligase-DpnI enzyme mix, which allowed for circularization of the PCR products and removal of template DNA. To construct the lentivirus transfer plasmids, OC1, OC2 and OC3 CDS regions were PCR amplified from pOC1(human), pOC2(human) and pOC3(human) templates using lvx-OC1-F and lvx-OC1-R, lvx-OC2-F and lvx-OC2-R, lvx-OC3-F and lvx-OC3-R primers, respectively, and inserted into pLVX-EF1α-IRES-mCherry (Addgene, here referred to as pLVXempty) between the XbaI and BamH1 sites. Primer sequences are listed in Supplementary Table 2.

## Transfection

Plasmids and RNA oligonucleotides (miRNA mimics and antagomirs) were transfected using Lipofectamine 3000 (Thermo Fisher) according to the manufacturer's instructions. In miRNA screening experiments, 0.2 μl of Lipofectamine 3000 reagent was used per well in 96-well plates. In ChIP experiments, 12 μl of Lipofectamine 3000 and 15 μl of P3000 reagents were used per plate in 100-mm plates. All the other transfection experiments were performed in 24-well plates in which 0.8 μl of Lipofectamine 3000 reagent was used per well for RNA transfection and 0.8 μl of Lipofectamine 3000 reagent plus 1 μl of P3000 were used per well for plasmid transfection. Synthetic miRNA mimics and antagomirs were purchased from Ribobio. Mimic sequences are listed in Supplementary Fig. 1b. miR-9 antagomir is completely complementary to miR-9. The NC antagomir sequence is 5′ CAGUACUUUUGUGUAGUACAAA 3′. Synthetic siRNAs were purchased from Sangon Biotech and their sequences are listed in Supplementary Table 3. The final concentrations of plasmids and RNA oligonucleotides are written in figure legends.

## Luciferase assays

For miRNA screening experiments, firefly luciferase activities were measured using a luciferase reporter gene assay kit (11401ES76, Yeasen Biotechnology). For dual-luciferase assays, a dual luciferase reporter gene assay kit (11402ES60, Yeasen Biotechnology) was used and renilla luciferase activities were normalized to firefly luciferase activities. A Varioskan Flash Spectral Scanning Multimode Reader (Thermo Scientific) was used for quantification of the signals.

## qPCR and RT−qPCR

To quantify miR-9 in cells or tissues, total RNA was purified using an Eastep Super total RNA extraction kit (catalogue no. LS1040m, Promega) following the manufacturer's protocol for retaining small RNA, reverse transcribed using a miRNA 1st Strand cDNA Synthesis Kit (catalogue no. MR101-01, Vazyme Biotech), and PCR amplified using a ChamQ Universal SYBR qPCR kit (catalogue no. Q711-02/03; Vazyme Biotech). The stem-loop primers and qPCR primers for miR-9 quantification were purchased from RiboBio. To quantify transcripts, RNA was isolated using an Eastep Super total RNA extraction kit (catalogue no.LS1040m, Promega), reverse transcribed using a HiScript II Q Select RT supermix kit (catalogue no. R233-01; Vazyme Biotech), and PCR amplified using a ChamQ Universal SYBR qPCR kit (catalogue no. Q711-02/03; Vazyme Biotech). The analyzed transcript levels were normalized to *GAPDH* transcript levels. To quantify viral genomes, DNA was isolated using a DNA isolation kit (catalogue no.DC102-01; Vazyme Biotech) and qPCR was conducted using the ChamQ Universal SYBR qPCR kit (catalogue no. Q711-02/03; Vazyme Biotech). Viral genome levels were normalized to mouse *Adipsin* gene levels. Serially diluted total DNA, total RNA, or synthetic miR-9 was used to generate standard curves. PCR primer sequences are listed in Supplementary Table 4.

## Lentivirus production

Four μg of lentivirus transfer plasmid, 7.1 μg of psPAX2 (Addgene) and 3.9 μg of pMD2.G (Addgene) were co-transfected into 293T cells in 100-mm plates using Lipofectamine 3000 (Thermo Fisher Scientific). At 16 h post-transfection, the supernatant was removed and replaced with fresh media. At 48 h post-transfection, the supernatant containing lentivirus was collected and stored at 4 °C and fresh media were added to the cells. At around 92 h post-transfection, another batch of supernatant was collected and combined with the first batch. The combined supernatant was centrifuged at 750 × *g* to remove cellular debris and further clarified with a 0.45-μM syringe filter. The lentivirus was concentrated by ultracentrifugation using a 50 Ti rotor (Beckman) at 110,000 × *g* for 90 min. The lentivirus pellet was resuspended in the media used for neuronal culture and stored at −80 °C. Lentivirus titers were determined by RT-qPCR using qLV-mcherry-F and qLV-mcherry-R primers.

## Construction of N2AmiR9 and N2A-C cell lines

Lentiviruses were produced using pTRIPZ empty vector or pTRIPZ-miR9 plasmid and added to 50% confluent Neuro-2a cells with 8 μg/ml hexadimethrine bromide (Sigma-Aldrich). Two days later, the supernatant was removed and replaced with fresh medium containing 1 μg/ml puromycin. Surviving cells were expanded in the presence of 1 μg/ml puromycin until stable cell lines were maintained.

## ChIP-qPCR

Cell monolayers in 100-mm plates were crosslinked in 1% formaldehyde for 10 min at room temperature, quenched with 0.125 M glycine for 5 min, washed twice with 5 ml of ice cold PBS. The cells were then scraped into 15 ml tubes with 5 ml of cold PBS and centrifuged at $750 \times g$ for 5 min at 4 °C. The cell pellet was lysed for 30 min on ice in 300 µl of 0.4% SDS lysis buffer (0.4% SDS, 10 mM of EDTA, 50 mM of Tris, pH8.0) and transferred into 1.5 ml Eppendorf tubes. The cell lysate was sonicated on ice using an SCIENTZ-IID Ultrasonic Homogenizer (SCIENTZ) for 10 cycles of 1 min each on the setting of 30 s ON, 30 s OFF, and 80 watts. Ten µl of sonicated cell lysate was used to determine the DNA concentration and size distribution and also served as the input sample. For this lysate fraction, crosslinks were reversed by incubating in 0.2 M of NaCl at 65 °C for over 3 h or overnight, followed by treatment with 0.2 mg/ml RNase (catalogue no. GE101-01; TransGen Biotech) for 1 h at 37 °C and then 0.5 mg/ml proteinase K (catalogue no. GE201-01; TransGen Biotech) at 58 °C for 1 h. DNA was purified using a HiPure Gel DNA Mini kit (catalogue no. D211102; Magen). DNA size distribution was analyzed by agarose gel electrophoresis. The procedure proceeded when optimal sizes of 200–500 bp were obtained. Each immunoprecipitation reaction was carried out by incubating a mixture containing 40 µl of sonicated cell lysate, 1 ml of ChIP dilution buffer (20 mM Tris-HCl (pH 8), 150 mM NaCl, 2 mM EDTA (pH 8), 1% NP-40, 0.1% sodium deoxycholate and 0.01% SDS, supplemented with protease inhibitor (1 tablet/50 ml, Roche, catalogue no. 4693132001)), 20 µl of Dynabeads Protein A (catalogue no. 10001; Invitrogen) and 2 µg of one of the following antibodies: anti-histone H3 (catalogue no. ab1791; Abcam); anti-H3K9me3 (catalogue no. ab8898; Abcam); anti-H3K27me3 (catalogue no. 9733; Cell Signaling Technology); anti-H3K27ac (catalogue no. ab4729; Abcam) or normal rabbit IgG (catalogue no. 12-370; Merck Millipore) at 4 °C with rotation overnight. The beads were sequentially washed once with cold low-salt wash buffer (150 mM NaCl, 20 mM Tris-HCl, pH8.0, 2 mM EDTA, 1% Triton X-100, 0.1% SDS, 1 mM phenylmethylsulfonyl fluoride (PMSF)), once with cold high salt wash buffer (same as low-salt wash buffer except that the NaCl concentration was 500 mM), once with cold LiCl wash buffer (50 mM HEPES, pH7.5, 250 mM LiCl, 1 mM EDTA, 1% NP-40, 0.7% sodium deoxycholate, 1 mM PMSF) and once with cold Tris-EDTA buffer (10 mM of Tris-HCl, pH 8.0, 1 mM of EDTA). The crosslinked DNA-protein complexes were eluted by incubation with 120 µl of elution buffer (1% SDS, 0.1 M of NaHCO₃) at 65 °C for 10 min. Crosslinks were reversed and DNA was extracted as above. The input and immunoprecipitated DNA samples were analyzed by qPCR using primers listed in Supplementary Table 4.

For analysis of OC2 enrichment, the steps upstream of immunoprecipitation were carried out as above. Each immunoprecipitation reaction was conducted in ChIP dilution buffer for 3 h at 4 °C using 150 µg of chromatin and 40 µl FLAG Affinity Gels (catalogue no. F2426; Merck Millipore). After the gels were washed as above, the crosslinked DNA-protein complexes were eluted by incubation with 90 µg of 3XFlag peptide (catalogue no.P9801; Beyotime) at 4 °C for 1 h. Crosslinks were reversed and DNA was extracted and quantified as above.

## Western blots

Western blotting was performed as described previously[63]. The following primary antibodies and dilutions were used: rabbit anti-OCT-1 antibody (1:10,000, catalogue no. ab178869; Abcam); rabbit anti-ONECUT2 antibody (1:2000, catalogue no. 21916-1-AP, Proteintech); mouse anti-FLAG antibody (1:2000, catalogue no. F1804, Sigma-Aldrich); rabbit anti-ONECUT1 antibody (1:500, catalogue no. A12774, ABclonal); rabbit anti-β-actin antibody (1:50,000, catalogue no. AC026, ABclonal); mouse anti-ICP4 antibody (1:5000, catalogue no. ab6514, Abcam); mouse anti-VP16 antibody (1:500, catalogue no. sc-7545, Santa Cruz). HRP-conjugated goat anti-mouse and goat anti-rabbit antibodies (1:2000, catalogue nos. 1030-05 and 4030-05;

SouthernBiotech). The validation information is provided in the Reporting Summary. Uncropped and unprocessed scans of western blots are provided in Source Data files.

## Mouse procedures

Mouse housing and experimental procedures were in compliance with national ethical guidelines and approved by the Laboratory Animal Welfare and Ethics Committee of Zhejiang University with an approval code of ZJU20220358. Institute for Cancer Research mice were purchased from the Shanghai Laboratory Animals Center. Mice were housed at ambient temperature (~23 °C) with low humidity in an air-conditioned room with 12 h light-dark cycles. Sex was not considered in this study. All experiments used male mice. For miR-9 quantification in different tissues, six-week-old mice were euthanized by cervical dislocation and tissues were removed and stored in the lysis buffer for RNA purification at −80 °C. Infection experiments were performed in a biosafety cabinet in an Animal Biosafety Level 2 laboratory. Six-week-old mice were anaesthetized by intraperitoneal injection of 0.4 ml of a mixture containing 4 mg/ml pentobarbital sodium (Solarbio) and 500 µg/ml xylazine hydrochloride (catalogue no. X1251; Sigma-Aldrich) in sterile saline. Three µl of PBS containing $2 \times 10^5$ pfu of HSV-1 was added onto each scarified cornea. For eye swab collection, mice were anaesthetized with 3% isoflurane (RWD Life Science) in oxygen with a flow rate of 0.5 ml/min using a V1 Table Top anaesthesia machine (Colonial Medical Supply). Cotton-tipped applicators were used to swab mouse eyes, and then suspended in 1 ml of cell culture medium. For quantification of infectious virus in TG, mice were euthanized by cervical dislocation and TG were harvested and then homogenized in cell culture medium before viral titration using plaque assays.

## Cultured neuron models of HSV-1 lytic infection, latent infection and reactivation

For lytic infection in primary mouse TG neurons, neurons were isolated from mouse TG and cultured in 48-well plates as described above. Three days later, the neurons were transduced with $10^8$ genome copies of lentivirus per well. Four days after transduction, neurons were infected with HSV-1. Supernatants were collected at the indicated times and titrated by plaque assays. Lentivirus transduction and HSV-1 lytic infection of differentiated human neurons were performed in a similar manner. To establish latency, two days after mouse TG neurons were cultured, they were treated with 100 µM ACV. The next day, neurons were infected with KOS (MOI = 20) in the presence of 100 µM ACV. The supernatants were replaced with fresh media containing 100 µM ACV at 2, 4 and 6 dpi. ACV was removed at 7 dpi by replacing the supernatant with ACV-free media. For the experiments using lentiviruses, neurons were transduced with $1.5 \times 10^8$ genome copies of lentivirus per well at 6 dpi in the presence of 100 µM ACV and 2 µg/ml hexadimethrine bromide (Sigma-Aldrich). On the next day, lentivirus and ACV were removed by medium replacement. Reactivation was triggered by adding DMSO, 0.2 µM TSA or 20 µM LY294002 with a DMSO concentration of 1:1000 three days after ACV removal. Then supernatants were collected every two days and assayed for the presence of infectious virus by evaluating cytopathic effects in Vero cells.

## Immunofluorescence assays

Cells were fixed in 4% paraformaldehyde for 10 min and washed with PBS. Cells were permeabilized in 0.1% Triton X-100 in PBS for 10 min and washed with PBS 3 times. Cells were blocked with 1% bovine serum albumin in PBS for 1 h at room temperature and then incubated with a mouse anti-TUJ1 antibody (1:1000, catalogue no. ab78078; Abcam) at 4 °C overnight and washed 3 times. Cells were incubated with goat anti-mouse IgG Alexa Fluor 647 (1:1000, catalogue no. ab150115; Abcam) for 1 h and then washed 4 times with PBS. Images were acquired on a Nikon Ti2-E fluorescence microscope with a 100×

magnification using the NIS Elements AR (Nikon) software. Images were analyzed using ImageJ v.1.52n (National Institutes of Health).

## Attachment and nuclear entry assays

Neuro-2a cells in 6-well plates were transfected with 400 ng pcDNA or pOC2ΔHOX per well using Lipofectamine 3000 (Thermo Fisher) for 42 h before infection. Cells were pre-chilled at 4 °C for 30 min before the addition of ICP0-null virus (MOI = 1) on ice. Then cells were incubated at 4 °C for 1 h with gentle rocking every 15 min. To analyze attachment, medium containing virus was removed and cells were washed on ice three times with cold PBS before being collected for viral genome analysis by qPCR. To analyze nuclear entry efficiencies, after the 1-h incubation at 4 °C, medium containing virus was removed and replaced with pre-warmed fresh medium before cells were incubated at 37 °C for 1 h. Cells (after the 37 °C incubation or, as a control, right after attachment at 4 °C) were washed once with cold PBS and scraped into 1 ml of cold PBS containing 0.5 mg/ml proteinase K (TransGen Biotech) and incubated at 4 °C for 2 h. Then, cells were washed three times with 1 ml of cold PBS each. Each wash was conducted by centrifugation at $1200 \times g$. for 3 min followed by supernatant removal and pellet resuspension. After the last wash, cells were resuspended in 1 ml of an ice-cold hypotonic buffer (10 mM of HEPES, pH 7.9, 1.5 mM of $MgCl_2$, 10 mM of KCl, 0.5 mM of DTT) and incubated on ice for 5 min. Cells were transferred into a 1-ml Dounce tissue grinder (catalogue no. 357538; DWK Life Sciences) and subjected to 15 strokes of homogenization using a tight pestle. Cells were transferred to another tube and centrifuged at $2400 \times g$ and 4 °C for 5 min. The supernatant was discarded and the pellet containing the nuclear fraction was collected for qPCR analysis of viral genomes.

## ChIP-seq

After following the ChIP protocols described above for purification of DNA from immunoprecipitated protein-DNA complexes, DNA libraries were constructed and sequenced by Tianjin Novogene Bioinformatic Technology Co., Ltd. Briefly, an NEBNext® Ultra™ II DNA Library Prep kit (catalogue no. E7645L; NEB) was used for library preparation. Libraries were analyzed for size distribution by an Agilent 5400 system (Agilent, USA) and quantified by qPCR. The quantified libraries were pooled and sequenced on an Illumina Novaseq600 platform to obtain raw data with 150 bp paired-end reads. Reads containing adapters and low-quality nucleotides were discarded to obtain clean reads using fastp (version 0.23.1). The genome of HSV-1 strain KOS (JQ673480.1) with one copy of the repeat sequences deleted[28] was concatenated to the mouse reference genome (mm10 assembly version) resulting in a combined genome named mm10+KOSnorepeat. The clean reads were aligned to the mm10+KOSnorepeat genome using Bowtie2 (version 2.3.4.3) with default parameters. Then the virus and mouse reads were splitted. DeepTools (verison 3.5.1) bamCoverage was used to calculate read coverages. PlotProfiles and ComputeMatrix were utilized to plot signals around the TSS. The peak calling algorithm MACS2 (version 2.2.7.1) was used to identify significant peaks representing DNA binding sites for mouse and virus separately. The peaks were annotated to known genomic features using ChIPseeker (version 1.32.1). Homer2 (version 4.11.1) findMotifsGenome.pl was used to identify enriched known and de novo motifs. Integrative Genomics Viewer (version 2.16.2) was used to visualize read coverages and identified peaks.

## RNA-seq

Neuro-2a cells were transfected with pOC2ΔHOX or pcDNA plasmid for 42 h and then infected with ICP0-null virus for 5 h (MOI = 1) before being harvested for RNA extraction using a ls1040 Eastep Super total RNA extraction kit (Promega). Library preparation and sequencing were performed by Beijing Genomics Institute. A Bigseq500 platform was used to generate 150 bp paired-end reads. The raw data were

filtered with SOAPnuke (v1.5.2) (https://github.com/BGI-flexlab/SOAPnuke). The clean data were mapped to the mm10+KOSnorepeat genome by Bowtie2 (version2.2.5) (https://bowtie-bio.sourceforge.net/bowtie2/index.shtml). Gene expression levels were calculated by RSEM (version1.3.1) (http://deweylab.github.io/RSEM/). Differential expression analysis was performed using DESeq2 (version1.4.5) (https://bioconductor.org/packages/release/bioc/html/DESeq2.html).

## Co-Immunoprecipitation

Neuro-2a cells in each 100-mm plate were transfected with 4 µg of an empty vector (pcDNA) or a plasmid expressing Flag-tagged OC2 (pOC2) for 36 h and then infected with KOS (MOI = 5) for 12 h. Cells were lysed in 1 ml of lysis buffer [50 mM HEPES-KOH (pH 7.4), 1% Triton X-100, 150 mM NaCl, 10% glycerol, 2 mM EDTA and Complete EDTA-free protease inhibitors (Roche, one tablet per 50 ml)] for 1 h. The lysates were incubated with 40 ml of Pierce Anti-Flag Magnetic Agarose (ThermoFisher, A36797) for 2 h. The agarose was washed three times with lysis buffer for a total of 1 h. The lysates (input) and immunoprecipitated samples (IP) were analyzed by Western blots using OC2, ICP4, and VP16 antibodies.

## ATAC-seq

Neuro-2a cells were transfected with pOC2ΔHOX or pcDNA plasmid for 40 h and then infected with ICP0-null virus for 2 h (MOI = 1). Fifty thousand infected Neuro-2a cells were collected and lysed with lysis buffer (10 mM Tris-HCl, pH 7.4, 10 mM NaCl, 3 mM $MgCl_2$, and 0.1% [vol/vol] Igepal CA-630) on ice for 7 min. Tagmentation and amplification were performed using a TruePrep DNA library Prep Kit V2 (TD501; Vazymes) and TruePrep Index Kit V2 for Illumina® (TD202; Vazyme). Libraries were sequenced on an Illumina Novaseq6000 platform at Tianjin Novogene Bioinformatic Technology Co., Ltd to generate 150 bp paired-end reads. Clean reads were obtained by removing reads containing adapters, reads containing ploy-N and low-quality reads from raw data using fastp (version 0.20.0) (https://github.com/OpenGene/fastp). Clean reads were aligned to the mm10+KOSnorepeat genome using BWA-MEM (https://github.com/lh3/bwa) and the resulting alignments were processed using sambamba (https://github.com/biod/sambamba) followed by filtering with SAMtools (https://github.com/samtools/samtools). The downstream analysis pipeline was adapted from a previous study[64].

## Statistical analyses

Statistical analyses were performed using Prism version 8.0.2 (Graph-Pad Software, www.graphpad.com). All measurements were taken from distinct samples. All multiple comparisons have been corrected before adjusted $P$ values are presented. The tests used are indicated in figure legends.

## Reporting summary

Further information on research design is available in the Nature Portfolio Reporting Summary linked to this article.

# Data availability

The ChIP-seq, RNA-seq and ATAC-seq data have been deposited to the Gene Expression Omnibus under accession numbers GSE241863, GSE241029 and GSE240221, respectively. The PAR-CLIP data from our previous study are available at Gene Expression Omnibus under accession number GSE127503. The HSV-1 KOS strain genome sequence and annotation were obtained from GenBank under accession number JQ673480.1. Mouse (mm10) genome sequences and annotations were downloaded from the UCSC Genome Browser (https://genome.ucsc.edu/cgi-bin/hgGateway?db=mm10). The crystal structure of OC1 DBDs complexed with DNA was obtained from Protein Data Bank under accession code 2D5V (https://www.rcsb.org/structure/2d5v). Micro-RNA expression profiles in different human tissues were obtained from

the TissueAtlas web server[29] (https://ccb-web.cs.uni-saarland.de/tissueatlas). MicroRNA target prediction was performed using datasets provided in TargetScan[30] (https://www.targetscan.org). Source data for this study are provided with this paper. Source data are provided with this paper.

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

## Acknowledgements

We thank George Church, David Leib and Jeremy Kamil for generously providing iNGN3 iPSCs, 64-GFP virus and PLAY2-I-SceIAphAI plasmid, respectively. We thank Xin Shen and Chun Guo at the Core Facility of Zhejiang University School of Medicine for expertise and instrument availability. We thank Jiaqi Li for her help with bioinformatics analysis. This work was supported by the National Key R & D Program of China (2023YFC2306700 to D.P.), the Natural Science Foundation of Zhejiang Province (LZ23H190001 to D.P.), the National Natural Science Foundation of China (82272322 to D.P.), the State Key Laboratory for Diagnosis and Treatment of Infectious Diseases, the First Affiliated Hospital, Zhejiang University School of Medicine (zz202313 to D.P.), Biota (to D.M.C.), and the National Institutes of Health (P01 AI098681 to D.M.C. and D.M.K.).

## Author contributions

D.P. conceived and supervised the study, performed experiments in cell culture, and prepared the draft of the manuscript. Y.D. performed most of the molecular cloning, animal experiments and cell culture experiments including the ChIP-qPCR and ChIP-seq analyses. Y.L. performed some molecular cloning and cellular experiments including RNA-seq and ATAC-seq analyses. S.C. helped with recombinant virus construction and animal experiments. D.P. with help from S.C. performed the experiments in primary neurons. Y.X. helped with Western blot and RT-qPCR analyses and data management. H.C. and S.Q performed miRNA screening. G.K.-M. generated and characterized the luc-HSV-1 virus as a screening tool; D.M.C. helped conceive and supervise that effort. H.S.O. and B.D. performed the experiments using differentiated human neurons and J.M.P. performed the statistical analysis of these studies; D.M.K. and D.M.C. helped supervise that effort. All authors revised and approved the manuscript.

## Competing interests

The authors declare no competing interests.
