## [Peer Review File · Nature Communications]

Reviewers' Comments:

Reviewer #1:

Remarks to the Author:

targeting cellular transcription activators Oct1 and OC members. Authors used Neuro-2A (mouse neuroblastoma) cells, mouse TG neurons and Human primary fibroblast cell line HFFs and 293T cell lines together with a mouse cornea infection model to test and verify their conclusions. Authors used many complementary techniques to support their findings including siRNA transfection-lentivirus transduction of shRNA-inducible cell line construction; characterization of heterochromatin marks on the viral genome with or without miR-9 or OC2 ectopic expression, mutational analysis of OC2 to identify required domain (CUT). Authors also generated recombinant virus mutant, HSV-1 Δ gC-luciferase, and used viral mutants including ICP0null, and HSV-2. The main novel findings of the papers are:

- 1- Mir-9 has a neuron specific anti-viral effect on HSV-1 productive replication, leading to a 10-fold reduction in the virus yield (neuro-2A cells)
- 2- miR-9 suppresses IE gene expression, increases heterochromatin on lytic genes (N-2A cells)
- 3- In vivo (and in TG neurons), lentivirus transduction of miR-9/ or HSV-1miR9 infection leads to reduced virus replication (not in the cornea) but in the TG.
- 4- In the in vitro latency model, introduction of miR-9 by lentivirus after the ACV removal, before the addition of reactivating stimuli, reduced the number of reactivation events.
- 5- Authors did not identify any single viral gene targeted by this miR, but they identified host cell factors Oct1 and onecut (OC) family members. They focused on OC2, since all the target sites were conserved among mouse and human sequences.
- 6- KO and KD conditions with Oct1 and OC2, respectively led to a reduction in HSV-1 yield (N-2A)
- 7- Overexpression of OC2 in N-2A and HFF (not in 293Ts) increased HSV-1 yield (wt and ICP0null). Similar findings in mouse TG infection with KOS strain.
- 8- OC proteins and miR-9 regulates HSV-2 replication similarly (in N-2A cells).
- 9- CUT domain (particularly the Q353 residue) was identified to be essential in the observed effects.
- 10- OC2 globally (non-specifically) stimulates the initial stage of HSV-1 transcriptional activation by decreasing heterochromatin.

Experiments and controls are well designed and clear, results support the conclusions of the paper. I have 2 minor concerns and a suggestion:

- 1-Authors used neuro-2A cells for many experiments to characterize the effect of miR-9 and OC2 overexpression, etc.. Although these cells may have neuron-like characteristics after differentiation, they are neuroblastoma cells from the brain and have CNS origin. HSV-1 replication and latency establishment are not well characterized in the CNS tissue and these neurons display a differential intrinsic immune response to HSV infections (compared to PNS neurons). Authors must describe how they differentiated the N-2A cells.
- 2- In Fig 2A, authors suggest that the lentivirus mediated miR-9 expression in TG neurons significantly reduced HSV-1 replication. Although the reduction is significant at 24 hpi, the yields are comparable at 48 hpi. This is not mentioned in the results section.
- 3- Authors suggest that OC2 displays widespread binding to the HSV-1 genome and globally stimulates gene transcription. Is this binding mediated by viral proteins, such as the viral transVP16? Did authors test a VP16null mutant? A co-IP with OC2 followed by a proteomic analysis would be informative to identify if there are any viral interactors.

Reviewer #2:

Remarks to the Author:

Mechanisms controlling herpes simplex virus (HSV-1) latency have been actively studied for decades and yet the molecular basis of the strict tropism for neurons, particularly those of the peripheral nervous system, remains poorly understood. Many features have been proposed including the unique cell morphology and highly-specialized physiology. Likewise, differences in transcription factor availability and in factors involved in formation of repressive chromatin have been evoked but without a clear smoking gun. Here the authors continue their already productive studies on the contribution of neuronal microRNAs (miRs) to the control of HSV-1 latency. Previously they identified miR-138 as a neuron-enriched cellular microRNA that antagonizes viral

replication by dampening the expression of the key viral activator ICP0 and host transcription factors OCT-1 and FOXC1. Using a simple but effective screen, they now identify miR-9 as another potent suppressor of HSV-1 replication and show that it also regulates OCT-1 as well as members of the OC family, which have not previously been linked to any alpha-herpesvirus. Indeed, they show that miR-9 and miR-138 collaborate in modulating the expression of multiple host and viral factors necessary for efficient expression of viral genes, uncovering an unrecognized regulatory network present in primary neurons but apparently, absent or much reduced in other cells.

The work is technically elegant and rigorous. It underscores the importance of microRNAs as a pre-existing determinant of cells that can support HSV-1 latency. However, there is an almost exclusive focus on control of the virus with less attention given to what this might mean outside of infection, which would broaden the appeal of the work and potentially leverage the virus to uncover important organismal biology. Is the network controlling something essential to neuronal physiology? Are uninfected neurons changed in some way when the miR-9/miR-138 network is perturbed by mimic expression? Uncovering neuronal properties that are reliant on this microRNA/transcription factor network might help to explain why the network is compromised in cells derived from tumors.

OTHER COMMENTS

Is there any evidence for cross-regulation of miR-138 by miR-9 or visa versa?

Figure 2. The logic of expressing additional miR-9 in neurons that already express it at already high physiological levels is not obvious. The effects are small. What is really learned from this? Can latency be achieved if ACV is left out or the length of exposure is reduced when TG neurons are infected with the HSVmiR9a/b viruses?

Reviewer #3:

Remarks to the Author:

In this manuscript, the authors sought to identify neuronal miRNAs that would influence HSV infection. To this, they transfected a range of miRs and found the strongest inhibition of viral gene expression with miR-9. As non-neuronal cells were not as greatly impacted by miR-9 exoression, and the target sites in late genes unable to explain effects on early viral genes, neuronal host factors were computationally determined and ONECUT transcription factors subsequently validated as miR-9 targets. Expression of these proteins was found to enhance viral genes expression and reactivation from latency in both culture and murine models. Interestingly, OC2 proteins were suggested to non-specifically bind viral DNA, contrary to host chromatin in the same assay. Expression of a range of OC2 mutants identified OC2 binds to viral DNA via the CUT domain while the HOX domain inhibited stimulatory effects. The OC2 HOX mutant was found to increase both viral RNA levels and chromatin accessibility.

Overall, the experiments are well performed and methods described. Each of the major conclusions is supported by multiple experiments and overall serve to identify a new pathway in which additional ONECUT family members impact early viral tranSCRIPTION, both in lytic infection and reactivation from latency.

There could be some additional discussion the nature of the associatino of OC2 with the viral genome, and why this appears different than that of the host. Have ONECUT proteins been found to interact with the major viral transcription factor ICP4, which also essentially binds the entire genome, or viral DNA by iPOND from the DeLuca lab?

A minor suggestion would be to italicize *in vivo/vitro* throughout the document and capitilize I when abbreviated for liter in the methods.

We thank the reviewers for taking time to review this manuscript and providing many helpful comments. We have addressed all points raised by the reviewers and included new data from the requested experiments. The point-by-point responses are written under the corresponding comments in bold letters. In addition, we added results showing effects of OC proteins on viral replication in human iPSC-derived neurons (Fig. 4E. The original Fig. 4E was moved to Fig. S5F) which further bolstered the story. We believe that the manuscript has been greatly improved.

Reviewer #1 (Remarks to the Author):

targeting cellular transcription activators Oct1 and OC members. Authors used Neuro-2A (mouse neuroblastoma) cells, mouse TG neurons and Human primary fibroblast cell line HFFs and 293T cell lines together with a mouse cornea infection model to test and verify their conclusions. Authors used many complementary techniques to support their findings including siRNA transfection-lentivirus transduction of shRNA-inducible cell line construction; characterization of heterochromatin marks on the viral genome with or without miR-9 or OC2 ectopic expression, mutational analysis of OC2 to identify required domain (CUT). Authors also generated recombinant virus mutant, HSV-1deltagC-luciferase, and used viral mutants including ICP0null, and HSV-2. The main novel findings of the papers are:

- 1- Mir-9 has a neuron specific anti-viral effect on HSV-1 productive replication, leading to a 10-fold reduction in the virus yield (neuro-2A cells)
- 2- miR-9 suppresses IE gene expression, increases heterochromatin on lytic genes (N-2A cells)
- 3- In vivo (and in TG neurons), lentivirus transduction of miR-9/ or HSV-1 miR9 infection leads to reduced virus replication (not in the cornea) but in the TG.
- 4- In the in vitro latency model, introduction of miR-9 by lentivirus after the ACV removal, before the addition of reactivating stimuli, reduced the number of reactivation events.
- 5- Authors did not identify any single viral gene targeted by this miR, but they identified host cell factors Oct1 and onecut (OC) family members. They focused on OC2, since all the target sites were conserved among mouse and human sequences.
- 6- KO and KD conditions with Oct1 and OC2, respectively led to a reduction in HSV-1 yield (N-2A)
- 7- Overexpression of OC2 in N-2A and HFF (not in 293Ts) increased HSV-1 yield (wt and ICP0null). Similar findings in mouse TG infection with KOS strain.
- 8- OC proteins and miR-9 regulates HSV-2 replication similarly (in N-2A cells).
- 9- CUT domain (particularly the Q353 residue) was identified to be essential in the observed effects.
- 10- OC2 globally (non-specifically) stimulates the initial stage of HSV-1 transcriptional activation by decreasing heterochromatin.

Experiments and controls are well designed and clear, results support the conclusions of the paper.

Response: We thank the reviewer for recognizing the novel findings from this work and the positive comments.

I have 2 minor concerns and a suggestion:

- 1-Authors used neuro-2A cells for many experiments to characterize the effect of miR-9 and OC2 overexpression, etc.. Although these cells may have neuron-like characteristics after differentiation,

they are neuroblastoma cells from the brain and have CNS origin. HSV-1 replication and latency establishment are not well characterized in the CNS tissue and these neurons display a differential intrinsic immune response to HSV infections (compared to PNS neurons). Authors must describe how they differentiated the N-2A cells.

Response: We did not differentiate Neuro-2a cells. To clarify this, we added the word “undifferentiated” the first time Neuro-2a cells appear. Undifferentiated cells are more convenient to use and easy to transfect. In our previous studies, results obtained in undifferentiated Neuro-2a cells could usually be recapitulated in primary neurons and/or mouse TG (e.g. Pan et al, 2014; Sun et al, 2021), although they are not suitable for establishing latency. In this study, after obtaining data from Neuro-2a cells, we used primary neurons isolated from mouse TG to validate the results (Fig. 2 for miR-9 and Fig. 4 for OC proteins), which we think is a superior model relative to differentiated Neuro-2a cells. Primary neurons were also used to investigate the effects on the latency-reactivation cycle. For miR-9, we also have some in vivo data from mouse trigeminal ganglia (Figs. 2B and 2C). Moreover, for OC proteins, we added data from iPSC-derived human neurons (added to Fig. 4G). Therefore, the mechanisms are functional in differentiated neurons as well as in undifferentiated Neuro-2a cells and appear to be conserved in CNS and PNS cells.

2- In Fig 2A, authors suggest that the lentivirus mediated miR-9 expression in TG neurons significantly reduced HSV-1 replication. Although the reduction is significant at 24 hpi, the yields are comparable at 48 hpi. This is not mentioned in the results section.

Response: We thank the reviewer for pointing this out. The insignificant difference at 48 h might be due to partial saturation of virus replication at later times. We re-wrote that sentence to read: “Transduced miR-9 significantly reduced HSV-1 yields at 24 hpi although the reduction became insignificant at 48 hpi.”

3- Authors suggest that OC2 displays widespread binding to the HSV-1 genome and globally stimulates gene transcription. Is this binding mediated by viral proteins, such as the viral transVP16? Did authors test a VP16null mutant? A co-IP with OC2 followed by a proteomic analysis would be informative to identify if there are any viral interactors.

Response: We thank the reviewer for raising the possibility that binding to viral genes might be mediated by viral proteins. We do not have a VP16null mutant virus. Regardless, we examined whether VP16 interacts with OC2 using co-IP but found no evidence of interaction (data added as Fig. S9).

We also thank the reviewer for the suggestion to use proteomic approaches to identify OC2 interactors. We already performed a preliminary experiment using this approach. We pulled down OC2 in HSV-1 infected Neuro-2a cells to identify interactors by mass spectrometry. The dataset did not identify any viral protein but identified over 100 potential host interactors. However, we feel that it is premature to put these data in this manuscript especially as the data are still preliminary and it would be beyond the scope of this study to explore the role of the numerous potential host interactors. If necessary, we would be happy to provide the dataset

on request, but only for the purposes of review.

Reviewer #2 (Remarks to the Author):

Mechanisms controlling herpes simplex virus (HSV-1) latency have been actively studied for decades and yet the molecular basis of the strict tropism for neurons, particularly those of the peripheral nervous system, remains poorly understood. Many features have been proposed including the unique cell morphology and highly-specialized physiology. Likewise, differences in transcription factor availability and in factors involved in formation of repressive chromatin have been evoked but without a clear smoking gun. Here the authors continue their already productive studies on the contribution of neuronal microRNAs (miRs) to the control of HSV-1 latency. Previously they identified miR-138 as a neuron-enriched cellular microRNA that antagonizes viral replication by dampening the expression of the key viral activator ICP0 and host transcription factors OCT-1 and FOXC1. Using a simple but effective screen, they now identify miR-9 as another potent suppressor of HSV-1 replication and show that it also regulates OCT-1 as well as members of the OC family, which have not previously been linked to any alpha-herpesvirus. Indeed, they show that miR-9 and miR-138 collaborate in modulating the expression of multiple host and viral factors necessary for efficient expression of viral genes, uncovering an unrecognized regulatory network present in primary neurons but apparently, absent or much reduced in other cells.

The work is technically elegant and rigorous. It underscores the importance of microRNAs as a pre-existing determinant of cells that can support HSV-1 latency. However, there is an almost exclusive focus on control of the virus with less attention given to what this might mean outside of infection, which would broaden the appeal of the work and potentially leverage the virus to uncover important organismal biology. Is the network controlling something essential to neuronal physiology? Are uninfected neurons changed in some way when the miR-9/miR-138 network is perturbed by mimic expression? Uncovering neuronal properties that are reliant on this microRNA/transcription factor network might help to explain why the network is compromised in cells derived from tumors.

Response: We thank the reviewer for the positive comments and for the suggestion that the network uncovered in this virology study could be relevant to organismal biology. We agree that the miRNA/transcription factor network might be important for neuronal physiology. However, we did not observe obvious changes in cell growth or morphology when Neuro-2a cells were transfected with miR-138/miR-9 mimic or transduced with a lentivirus expressing miR-138 or miR-9, which means that their roles are subtle. Elucidating the subtle roles of these miRNAs in neuronal physiology would be beyond the scope of this study. However, to broaden the appeal of this work, we added the following sentences to the end of the last paragraph of Discussion: “Interestingly, the neuronal miRNAs have an enriched function in targeting transcription factors (Fig. S1C). The miRNA-transcription factor network, which has yet to be thoroughly explored, might be essential for neuronal physiology, and HSV appears to exploit the host network to help achieve persistence in neurons.”

OTHER COMMENTS

Is there any evidence for cross-regulation of miR-138 by miR-9 or visa versa?

Response: We have performed this experiment and found no cross-regulation between miR-138 and miR-9. This gives us another reason to focus on the mRNA targets of miR-9. We have added the results to Fig. S4A.

Figure 2. The logic of expressing additional miR-9 in neurons that already express it at already high physiological levels is not obvious. The effects are small. What is really learned from this?

Response: The logic of expressing additional miR-9 in neurons is to reveal its regulatory potential especially when its endogenous functions might be masked by redundant mechanisms. Not all effects are small. For example, the effects of lentivirus expressing miR-9 on reactivation were quite robust (Fig. 2E). We had difficulties in inhibiting miR-9 functions in neurons. Recently we achieved decent inhibition and showed that an antisense sequence can at least partially de-repress OC2 (Fig. S2F). By using that antisense sequence, we assessed the effects of inhibition of miR-9. However, we did not see significant effects on viral replication and reactivation. Although these results are negative, they are consistent with presence of redundant HSV-1 silencing mechanisms in neurons. Therefore, we added these data (Fig. 2A and 2E) and explained that this might be due to the presence of redundant mechanisms (e.g., miR-138).

Can latency be achieved if ACV is left out or the length of exposure is reduced when TG neurons are infected with the HSVmiR9a/b viruses?

Response: The recombinant HSV-1 viruses were generated to conduct in vivo experiments for which we do not have a choice of introducing lentivirus into TG. The recombinant viruses are not ideal to use in neurons since expression from recombinant HSV-1 is usually less robust than that from lentivirus. But in mouse TG, especially when viral replication was low, they sometimes show significant differences from control virus (like at 7 days post-infection in Figs. 2B and 2C). Regardless, we performed the experiment of leaving out ACV to basically let the viruses undergo lytic infection. The results showed similar replication of HSV1control, HSV1miR9a and HSV1miR9b as shown in the figure below. All neurons died around 4 days post-infection so latency was not achieved for any virus even though a low MOI of 0.1 was used. Given such results we do not think it is necessary to perform the trickier experiment of reducing the length of ACV exposure. Because we have already demonstrated the effects of miR-9 overexpression in neuronal culture using lentivirus transduction (Figs 2A), the negative results from recombinant viruses could well be due to insufficient miR-9 overexpression. Therefore, the figure below is not added to the manuscript.

Reviewer #3 (Remarks to the Author):

In this manuscript, the authors sought to identify neuronal miRNAs that would influence HSV infection. To this, they transfected a range of miRs and found the strongest inhibition of viral gene expression with miR-9. As non-neuronal cells were not as greatly impacted by miR-9 exoexpression, and the target sites in late genes unable to explain effects on early viral genes, neuronal host factors were computationally determined and ONECUT transcription factors subsequently validated as miR-9 targets. Expression of these proteins was found to enhance viral genes expression and reactivation from latency in both culture and murine models. Interestingly, OC2 proteins were suggested to non-specifically bind viral DNA, contrary to host chromatin in the same assay. Expression of a range of OC2 mutants identified OC2 binds to viral DNA via the CUT domain while the HOX domain inhibited stimulatory effects. The OC2 HOX mutant was found to increase both viral RNA levels and chromatin accessibility.

Overall, the experiments are well performed and methods described. Each of the major conclusions is supported by multiple experiments and overall serve to identify a new pathway in which additional ONECUT family members impact early viral transcription, both in lytic infection and reactivation from latency.

Response: We thank the reviewer for these positive comments.

There could be some additional discussion the nature of the associatino of OC2 with the viral genome, and why this appears different than that of the host.

Response: We modified and extended the following sentences in Discussion: "The global impact on viral gene expression and chromatin is consistent with widespread binding to viral DNA with little sequence specificity. Like other sequence-specific DNA binding proteins, OC proteins doubtless have some weaker, basal nonspecific DNA binding activity, which we speculate would depend on hydrogen bonding between the CUT domain and DNA backbone (Fig. 5C). HSV-1 chromatin, at least during lytic infection, is known to be generally less compacted than host chromatin⁵⁵. The relatively open viral chromatin might allow OC proteins to bind in a nonspecific and widespread manner to further increase chromatin accessibility. In contrast, the largely compacted host chromatin permits OC proteins access to

only certain genomic regions and stable binding to such regions might require sequence specific interactions.

Have ONECUT proteins been found to interact with the major viral transcription factor ICP4, which also essentially binds the entire genome, or viral DNA by iPOND from the DeLuca lab?

Response: We thank the reviewer for raising the possibility that OC proteins interact with ICP4, a viral protein with widespread binding to the viral genome. To address this possibility, we performed co-IP in HSV-1 infected Neuro-2a cells, but found no evidence that OC2 interacts with ICP4. The data is added to Fig. S9. Regarding whether OC proteins interact with viral DNA by iPOND from the DeLuca lab, we again appreciate the suggestion but our laboratories have not established the iPOND method. Since we have the ChIP-seq data which already demonstrate that OC proteins interact with viral DNA, we think that iPOND is not necessary for this manuscript. We will keep this technique in mind for future studies.

A minor suggestion would be to italicize *in vivo/vitro* throughout the document and capitalize *l* when abbreviated for liter in the methods.

Response: We have italicized *in vivo/vitro* throughout the manuscript. Regarding whether to capitalize the *L* for liter, different journals might have different requirements, so we will leave it to the journal editors.

Reviewers' Comments:

Reviewer #1:

Remarks to the Author:

Authors have adequately addressed the questions and issues raised by the reviewers in their revised manuscript.

Reviewer #2:

Remarks to the Author:

The initial reviews were favorable and for the most part, asked either for clarifications or for deeper dives into the data. The authors have responded well providing a significant number of new experiments, selecting the positive findings for inclusion in the revised manuscript. The inclusion of new studies using human iPSC neurons is a nice addition.

Reviewer #3:

Remarks to the Author:

In this resubmission, the authors have added additional text and experimental data, as well as addressing further concerns from the previous version. Overall, this manuscript uses a range of experimental techniques to link ONECUT protein family members as a factor regulating the neuronal latency of HSV, a virus that goes lytic in essentially every other cell type. It provides a significant direction for further research and I have no further concerns.